# A tin fluoride-free, efficient and durable tin-lead perovskite solar cell

Haobo Yuan[1,3], Wenxiao Zhang ®[1,3] ✉, Feng Wang ®[2], Jianhong Xu[1], Yuyang Hu[1], Xuemin Guo[1], Yunfei Li[1], Bo Feng[1], Zhengbo Cui[1], Wen Li[1], Sheng Fu[1], Xiaodong Li ®[1], Feng Gao ®[2] ✉ & Junfeng Fang ®[1] ✉

The photo-thermal stability of tin-lead perovskite solar cells remains a major challenge. $SnF_2$ is commonly used to inhibit $Sn^{2+}$ oxidation and reduce hole density, however, the stability of devices remains poor. Here, we found that the poor stability partially results from an adverse effect of $SnF_2$, which reacts with formamidine iodide during photo-thermal treatments. This reaction leads to degradation of perovskite and release of hydrofluoric acid, which corrodes electrodes. To address this issue, we develop a strategy that combines lead powder in precursor with $PbF_2$ post-treatment, replacing the role of $SnF_2$ as in film formation and surface defect passivation, respectively. The d-electron polarization in $Pb^{2+}$ strengthens its bond with $F^-$, making it react inert to perovskite. In this work, the efficiency of $SnF_2$-free devices increases from 16.43% to 24.07%. The cells retain 60% of their initial efficiency after 550 hours operating at 85 °C under maximum power point.

Tin-lead perovskite solar cells (Sn-Pb PSCs) are recognized for their potential to achieve the highest Shockley-Queisser limit among single-junction solar cells, which has sparked considerable research interest[1]. The certified efficiency of Sn-Pb PSCs has indeed surpassed 24%, and it is conceivable that they may match or even exceed the performance of lead-based PSCs in the future[2]. However, the stability of Sn-Pb PSCs remains the most significant challenge, particularly under light and thermal conditions. This is primarily attributed to the oxidation of stannous ion ($Sn^{2+}$), Sn vacancy defects, iodine (I)-related defects, phase separation, and interfacial degradation reactions[3–7].

All high-efficiency Sn-containing PSCs rely on the incorporation of $SnF_2$ as an essential additive. From the earliest studies to the latest reports, Sn-Pb devices fabricated without $SnF_2$ have never exceeded 15% efficiency-originally below 10% and a viable alternative to $SnF_2$ remains elusive[8–13]. $SnF_2$ plays a crucial role in inhibiting the oxidation of $Sn^{2+}$ and the formation of tin vacancies ($V_{Sn}$), thereby reducing the background hole density and mitigating p-type doping. This effect can be attributed to the Sn compensating effect and the stronger Sn-F bond compared to Sn-I bond[8–17]. However, the roles of $SnF_2$ appears to be more complex. Studies have shown that a large amount of $SnF_2$ tends to aggregate preferentially on the bottom surface, followed by the top surface, with very little present in the bulk, which is an uncontrollable process[11,18]. This is possibly due to the large mismatch between Sn-I and Sn-F bonds partially hinders the assimilation of $SnF_2$ into the bulk. On the top surface, moderate $F^-$ ions are particularly well-suited to bind strongly with Sn atoms, thereby increasing the formation energy of $V_{Sn}$ defects[19,20]. Nevertheless, the enrichment of $SnF_2$ at the bottom surface (the hole transport interface) would cause downward band bending and create a hole transport barrier due to the de-p-doping function of $SnF_2$, thereby restricting hole transport in p-i-n structure of Sn-containing PSCs. Despite numerous efforts that have significantly enhanced stability, performance under bias, light, thermal, or combined stress conditions remains limited, even when cells are subjected to inert conditions or hermetically sealed to prevent moisture and oxygen ingress[21–30].

Herein, we find that $SnF_2$ can react with formamidine iodide (FAI) component in the perovskite film during thermal treatments, a process accelerated by light soaking, which disrupts the perovskite structure. Moreover, volatile byproducts like hydrofluoric acid (HF) can erode ITO and metal electrodes, and the inhibition of $Sn^{2+}$ oxidation becomes

[1]School of Physics and Electronic Science, Engineering Research Center of Nanophotonics & Advanced Instrument, Ministry of Education, East China Normal University, Shanghai, China. [2]Department of Physics, Chemistry and Biology (IFM), Linköping University, Linköping, Sweden. [3]These authors contributed equally: Haobo Yuan, Wenxiao Zhang ✉e-mail: wxzhang@phy.ecnu.edu.cn; feng.gao@liu.se; jffang@phy.ecnu.edu.cn

ineffective once F⁻ ions are depleted (Fig. 1). To avoid the adverse effects of SnF₂ on stability and hole transport, we replace SnF₂ additive with lead powders—known for its antioxidant and crystallization-regulating effects as reported in our previous work—to remove Sn⁴⁺ from the precursor, combined with a PbF₂ post-treatment to passivate surface defects[26]. The polarization of d-electrons in Pb²⁺ enhances its bonding with F⁻ ions, reducing its likelihood to react with FAI. Meanwhile, Pb²⁺ could fill $V_{Sn}$ and F⁻ could suppress the formation of $V_{Sn}$ by forming Sn-F bond. By employing this strategy, we increase the efficiency of SnF₂-free Sn-Pb PSCs from 16.43% to 24.07% with improved photo-thermal stability, where the cells remain 60% of their initial efficiency after continuous operation at 85 °C under maximum power point (MPP) conditions for 550 h.

## Results

### Side reaction between Sn-Pb perovskite and SnF₂ additive

Halide perovskite ($Cs_{0.2}FA_{0.8}Sn_{0.5}Pb_{0.5}I_3$ in this work) forms perfectly through the reaction between monovalent halides (CsI and FAI) and metal halides (SnI₂ and PbI₂) when the precursor is prepared in accordance with the stoichiometric ratio of the molecular formula. However, in the presence of 10 mol% SnF₂ relative to SnI₂, the spontaneous reaction between FAI with SnF₂ (2FAI + SnF₂ → SnI₂ + 2FA + 2HF) disrupts the stoichiometry of the precursor and interferes with the formation of perfect crystals. Thermogravimetric-differential scanning calorimetry (TGA-DSC) is performed on the FAI and FAI+SnF₂ powders (with a molar ratio of 2:1) in N₂ atmosphere from room temperature to 600 °C (Fig. 2a and Supplementary Fig. 1). FAI powder begins to decompose at 235 °C[31], while FAI+SnF₂ powders start to decompose at temperature below 100 °C, accompanied by two distinct exothermic peaks at 60 and 102 °C (Supplementary Fig. 1). Meanwhile, the final weight loss of FAI+SnF₂ powders is 25.9% matching the calculated weight fraction of FA and HF in FAI+SnF₂ powders (25.9%). Thermogravimetric-mass spectrometry (TGA-MS) analysis of $FASn_{0.5}Pb_{0.5}I_3$ powders, both with and without the addition of 10 mol% SnF₂, also reveals that SnF₂ significantly accelerates the decomposition process of perovskite. This is accompanied by a noticeable release of HF and FA, which is more pronounced compared to the decomposition of the perovskite without SnF₂ (Fig. 2b). Nuclear Magnetic Resonance (NMR) characterizations also verify this reaction. In the ¹H NMR spectrum of FAI, the primary characteristic signals of FA⁺ are observed at 7.8 ppm (assigned to the CH group) and 8.7 ppm (assigned to the NH₂ group) (Supplementary Fig. 2). Upon the addition of SnF₂ to the FAI solution, a noticeable splitting pattern of the NH₂ signal is observed, indicating a strong interaction between the F⁻ ions in

SnF₂ and the NH₂ group in FA⁺. After aging at 85 °C for 3 h, a new signal peak of triazine merges at 9.3 ppm, indicating the deprotonation of FA⁺. Scanning electron microscopy (SEM) of perovskite film shows cracks in the presence of SnF₂ after aging at 85 °C with light soaking for 200 h, while nearly no change is observed in its absence, intuitively reflecting the detrimental effect of SnF₂ on film stability (Fig. 2c, d and Supplementary Fig. 3a, b). This accelerated degradation effect of SnF₂ is further investigated using X-ray photoelectron spectroscopy (XPS). The N 1s spectra reveals two components in both fresh and aged perovskite, corresponding to FA⁺ at higher binding energy and reaction by-products (FA) at lower binding energy (Fig. 2e, f and Supplementary Fig. 3c, d)[32]. In the presence of SnF₂, the fraction of FA is 22.5% in the fresh perovskite film and increases to 32% after aging at 85 °C with light soaking for 200 h. In comparison, the reaction by-products fraction of SnF₂-free perovskite is only 6.7% in the fresh state and 8.6% after aging, showing little increase. Meanwhile, the disappearance of the F 1s peak after this aging process is consistent with the reaction between of FAI and SnF₂ (Supplementary Fig. 4).

### Instability issues in Sn-Pb PSCs caused by SnF₂ additive

In addition to the reaction between SnF₂ and FAI, the migration of F⁻ and diffusion of HF through the whole device simultaneously corrode all the functional layers, including anode and cathode during photo-thermal degradation process. Time-of-flight secondary ion mass spectrometry (TOF-SIMS) analysis of a complete device after continuous operating at 85 °C under MPP for 200 h shows the distribution of F⁻ from the Cu cathode to ITO anode (Fig. 3a and Supplementary Fig. 5). This result is driven by the spontaneous reactions that generate HF, which in turn damages the perovskite crystal structure and accelerates ion migration. The SnF₂-free device, subjected to a PbF₂ post-treatment, exhibits minimal diffusion of F⁻ ions. This observation indicates the low reactivity between PbF₂ and FAI, which will be discussed in detail in the following section. The activation energy ($E_A$) for ion migration decreases from 0.336 eV to 0.284 eV after introducing SnF₂ into perovskite before aging (Supplementary Fig. 6). These processes mutually reinforce each other during thermal or light aging, ultimately leading to a vicious cycle of degradation within the device. After aging at 85 °C with light soaking, the Fermi level downshifts at the surface, further increasing the electron transfer barrier (Fig. 3b and Supplementary Fig. 7a). This degradation process alters the energy levels of perovskite, particularly at the surface (Supplementary Fig. 8a), while the bulk perovskite remains largely unchanged (Supplementary Fig. 8b). In contrast, although the Fermi level in SnF₂-free perovskite

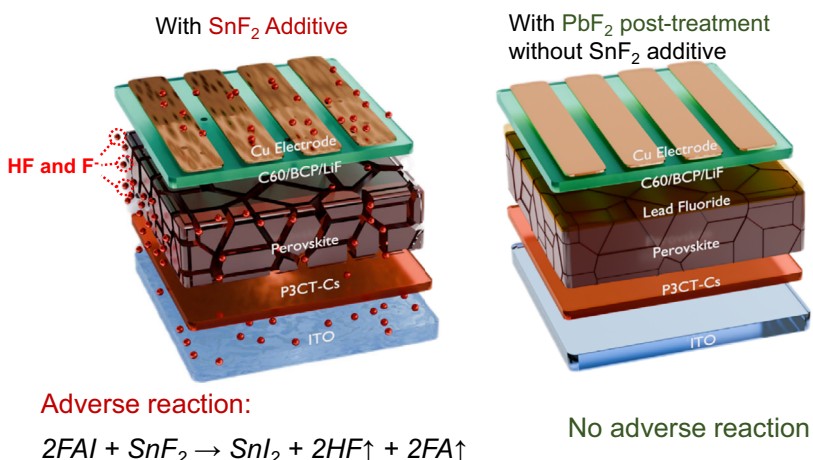

**Sn-Pb PSCs Operated under 85℃ MPP Condition**

With SnF₂ Additive

HF and F⁻

Adverse reaction:

$2FAI + SnF_2 \rightarrow SnI_2 + 2HF\uparrow + 2FA\uparrow$

With PbF₂ post-treatment without SnF₂ additive

No adverse reaction

**Fig. 1 | Comparison of photothermally degraded devices.** Scheme of the Sn-Pb PSCs with SnF₂ additive versus those without SnF₂ but with PbF₂ surface post-treatment after photo-thermal degradation.

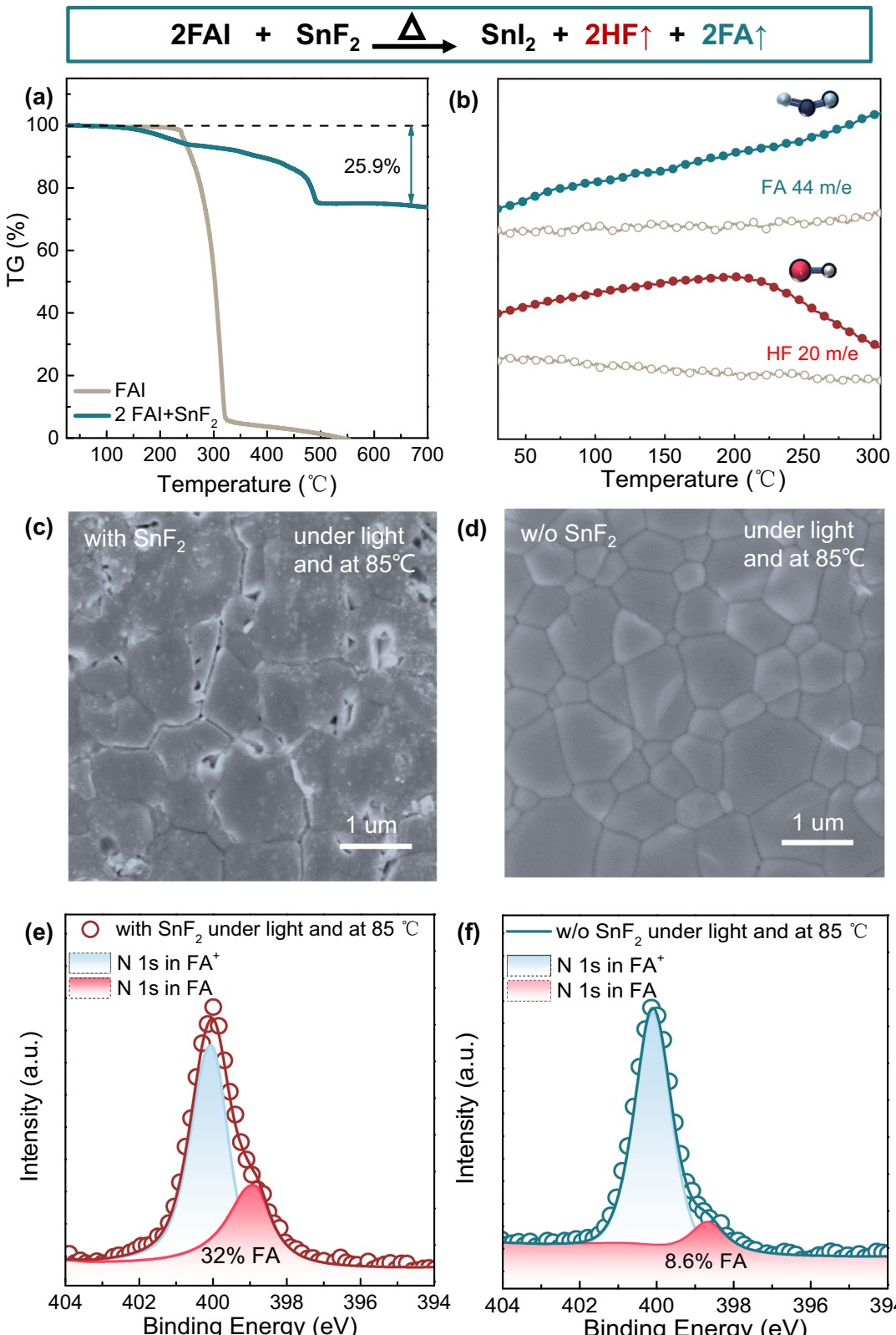

**Fig. 2 | The reaction between SnF₂ and FAI in perovskite film. a** TGA heating curves expressed as weight % as a function of applied temperature. **b** TGA-MS results of FASn$_{0.5}$Pb$_{0.5}$I$_3$ perovskite precursor powders with (solid circle) and without (hollow circle) SnF₂. **c, d** SEM spectra of Sn-Pb perovskite film aging at 85 °C with light soaking for 200 h. **e, f** N *1s* XPS spectra of Sn-Pb perovskite film aging at 85 °C with light soaking for 200 h.

remains relatively stable, both fresh and aged perovskite exhibit a high electron transport barrier, which is one of the reasons for the low performance of SnF₂-free devices and highlights the need for further optimization (Supplementary Figs. 7b and 8c, d). We investigated the

real-time variations in the electrical conductivity of Cu electrodes under continuous light soaking at room temperature, a condition known to accelerate ion migration, as illustrated in the inset of Fig. 3c[33]. The conductivity of Cu electrodes in SnF₂-containg device drops to

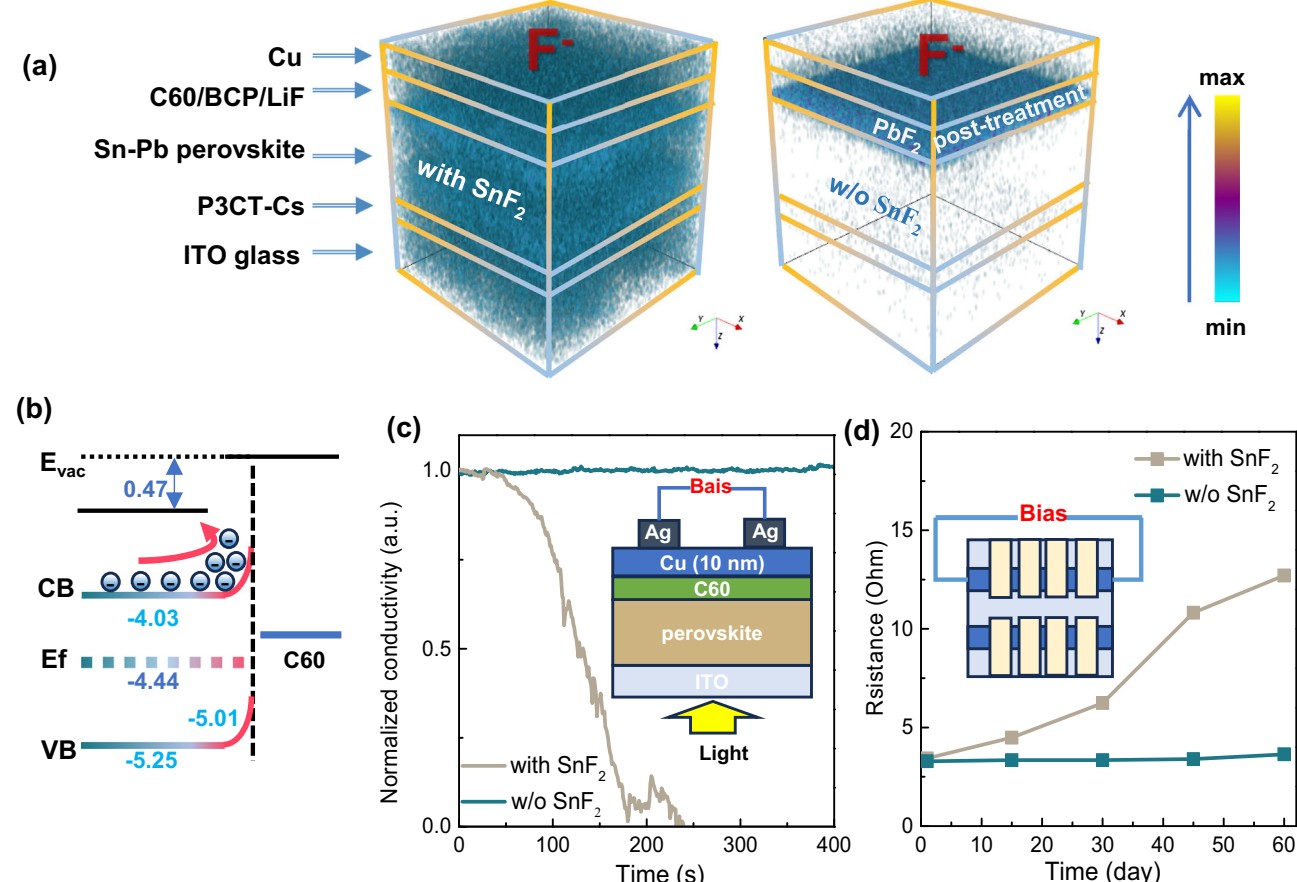

**Fig. 3 | The impact of SnF$_2$ on the stability of each functional layer in the device.** **a** TOF-SIMS 3D tomography results for a Sn-Pb PSC with SnF$_2$ additive after aging at 85 °C under MPP condition for 200 h. **b** The energy band of perovskite/C60 interface with SnF$_2$ additive after aging at 85 °C under MPP condition for 200 h. **c** The real-time changes in the electrical conductivity of Cu electrode in Sn-Pb perovskite with and without SnF$_2$ additive during light soaking. **d** The resistance of ITO in Sn-Pb perovskite with and without SnF$_2$ additive aging at room temperature.

zero even within 200 s, while no decline is observed in SnF$_2$-free devices. Additionally, we compare the resistance of ITO in these two types of devices by aging them at room temperature and measuring their resistance periodically as illustrated in the inset of Fig. 3d. The resistance of ITO in the SnF$_2$-containing device increases from 3 to 13 Ohm after 60 days, while that of SnF$_2$-free device shows no increase. After 48 h immersion in a 0.1 M SnF$_2$/FAI mixed solution followed by DMF rinsing, the pronounced SEM transformation of ITO from a dense, continuous surface to one marked by conspicuous inter-particle gaps, provides direct evidence that the reaction products of SnF$_2$ and FAI compromise the structural integrity of the ITO substrate (Supplementary Fig. 9). The changes of both Cu and ITO electrodes even at room temperature or without light soaking prove the strong corrodibility of migrated ion and reaction products.

## Effective and stable alternative strategy to SnF$_2$ additive

To avoid the adverse effects of SnF$_2$ on stability and hole transport at the bottom of perovskite film, its positive role during film-forming process and on top surface defect passivation need to be strategically replaced. Toward this goal, we first investigate the use of lead powder and tin powder as a sacrificial agent to inhibit the oxidation of Sn$^{2+}$ in the precursor and to regulate crystallization[26]. Pb powder react fully with FAI, releasing extra FA that regulate the crystallization growth (Supplementary Figs. 10 and 11) and suppress the formation of V$_{Sn}$ (Supplementary Fig. 12) in the perovskite films. However, the device efficiency based on the addition of lead powder is lower than that of SnF$_2$ based devices, confirming the role of SnF$_2$ is not limited to as the

sacrificial agents. Giving the distribution of SnF$_2$ in resulted perovskite films both at the surface and the bottom of film, where the bottom SnF$_2$ would create a hole transport barrier due to the de-p-doping function of SnF$_2$ (Supplementary Fig. 13), we further introduce a similar fluoride, PbF$_2$, as a passivation layer (Fig. 4a). After PbF$_2$ post-treatment, SEM-EDS (energy-dispersive X-ray spectroscopy) shows the surface Sn/Pb ratio falls from 1.17 to 0.78 (Supplementary Fig. 14 and Supplementary Table 1). Compared with Sn$^{2+}$, Pb$^{2+}$ forms stronger Pb-I bonds that are harder to break, effectively suppressing V$_{Sn}$ defects[34]. PbF$_2$ is expected to have a low reactivity with FAI, as the presence of more d-electrons in Pb$^{2+}$ could enhance its polarization effects and strengthen its ability to bind extranuclear electrons and electrophilic F$^-$, thus increasing the reaction barrier with FAI in perovskite structures compared to those in SnF$_2$. This can be confirmed by the TGA results, where FAI+PbF$_2$ powders shows higher decomposition temperature (over 100 °C) than that of FAI+SnF$_2$ powders (Fig. 4b). FAI+PbF$_2$ powders exhibit a 3.1% weight loss at around 164 °C (Supplementary Fig. 15), which correlates well with the calculated weight fraction of HF in FAI+PbF$_2$ powders (3.4%). The $^1$H NMR of FAI-PbF$_2$ blend shows no change in the NH$_2$ signal, or any new peaks after aging at 85 °C for 3 h, confirming PbF$_2$ is inert toward FAI (Supplementary Fig. 16). A more intuitive observation is that mixing FAI and SnF$_2$, whether stirred at room temperature or heated at 100 °C, results in the formation of black substances (FASnI$_3$). In contrast, mixing FAI with PbF$_2$ shows no color change even after heating at 100 °C (inset in Supplementary Fig. 17). The X-ray diffraction (XRD) patterns of the films formed from these mixtures upon heating at 100 °C are shown in Supplementary

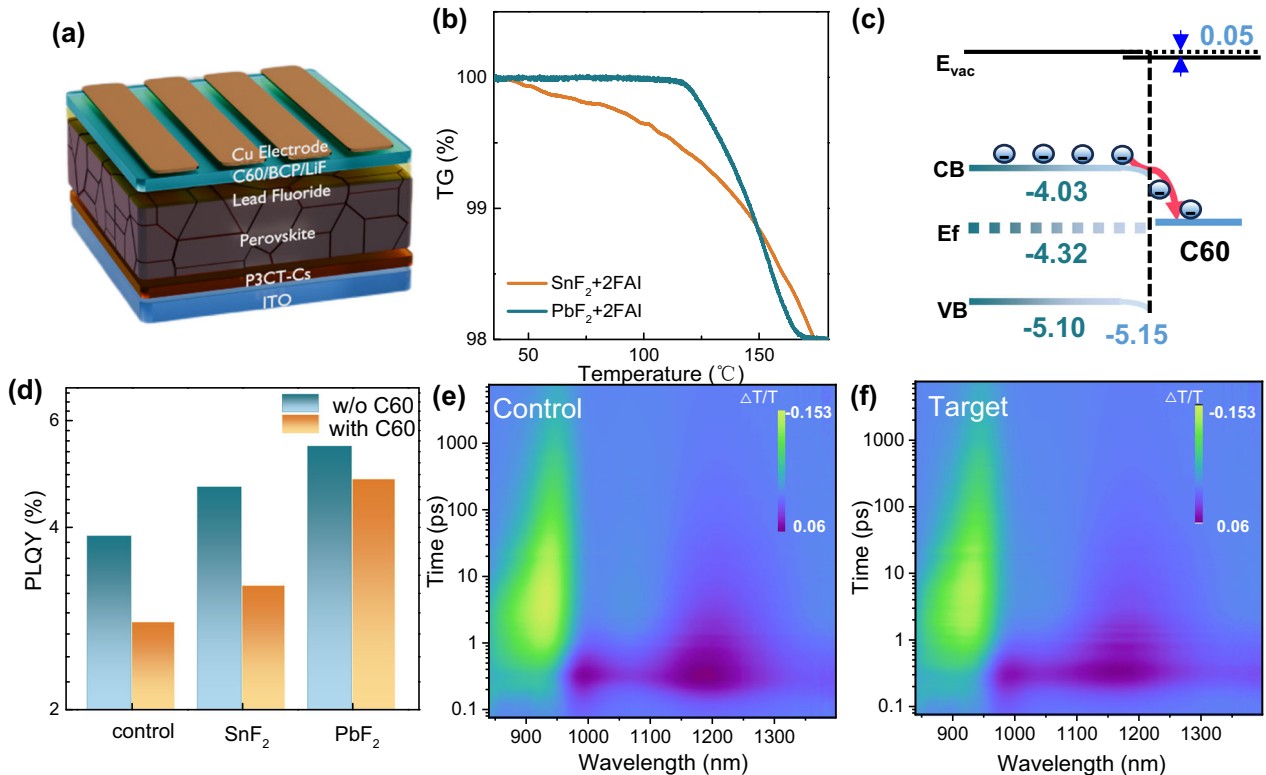

**Fig. 4 | The function of PbF₂ post-treatment on stability and charge dynamics.**
**a** Schematic of the device structure. **b** TGA heating curves of FAI+PbF₂ and FAI
+SnF₂ powders expressed as weight % as a function of applied temperature. **c** The
energy band of target perovskite/C60 interface regardless of fresh or after 200 h
aging at 85 °C with light soaking. **d** PLQY of the control, target and SnF₂ doped Sn-
Pb perovskite films, with and without C60 deposition. TA spectra for the control (**e**)
and target (**f**) perovskite films.

Fig. 17. Notably, the FAI+SnF₂ film exhibits distinct peaks characteristic
of the FASnI₃ perovskite phase, indicating the formation of SnI₂
resulting from the reaction between FAI and SnF₂. In contrast, the XRD
peaks of the FAI+PbF₂ film do not correspond to the FAPbI₃ phase but
rather indicate the presence of a complex between FAI and PbF₂,
confirming no reaction occurs even at 100 °C. SEM morphology of
perovskite films with PbF₂ post-treatment also show nearly no change
after aging at 85 °C with light soaking for 200 h (Supplementary
Fig. 18). The reaction between PbF₂ and FAI in actual perovskite films
has also been confirmed through XPS analysis. The fraction of reaction
by-products in PbF₂ post-treated perovskite (target) is minimal, at only
7.0% before aging and 8.8% after aging at 85 °C with light soaking for
200 h (Supplementary Fig. 19). These values are comparable to those
of the SnF₂-free film (control), as illustrated in Fig. 1d and Supple-
mentary Fig. 2b. PbF₂'s inertness toward FAI also suppresses FA
vacancies ($V_{FA}$). Both $V_{FA}$ and $V_{Sn}$ are p-type defects that pin the Fermi
level near the valence band edge. Dense $V_{Sn}$ leaves the lattice iodine-
rich, and $V_{FA}$ lowers the I⁻ ion migration barrier (Supplementary Fig. 6),
spawning more iodine-based p-type defects ($I_{Sn}$ and $I_{FA}$) that reinforce
one another[35]. Their accumulation bends the bands upward at the
perovskite/C60 interface. By mitigating these defects, PbF₂ post-
treatment establishes a stable, uniform energy alignment at the per-
ovskite/C60 interface (Fig. 4c, Supplementary Figs. 20–22).

To access the quality of interface contact and the extent of non-
radiative recombination in PbF₂ post-treated perovskite films, we
investigated their photo-luminescence quantum yield (PLQY). The
PLQY of the control perovskite film is only 3.875%, which increases to
4.675% and 5.454% following SnF₂ additive and PbF₂ post-treatment,
respectively. Upon creating a perovskite/C60 interface, the PLQY of the
PbF₂ post-treatment device retains 88%, while the control and SnF₂
additive devices retain only 72% and 68% PLQY, respectively (Fig. 4d).

These findings indicate that the PbF₂ post-treatment approach is
effective in reducing non-radiative recombination loss at this
interface[36]. Femtosecond transient absorption (fs-TA) spectroscopy is
conducted to probe the carrier transport dynamics. Figure 4e, f pre-
sents pseudo-color images of the TA spectra ($\Delta T/T$) for control and
target samples (glass/perovskite/(PbF₂)/C60). The target sample shows
faster ground-state bleach (GSB) (at 926 nm) and hot-exciton absorp-
tion (at 994 and 1170 nm) signal decay than the control sample (Sup-
plementary Fig. 23 and Supplementary Table 2). Meanwhile, the carrier
transport dynamics in target sample is comparable to those in samples
with SnF₂ additive (Supplementary Fig. 23 and Supplementary Table 2),
demonstrating the effectiveness of this alternative method. The shorter
carrier lifetime on microsecond timescale of target perovskite film
(5.28 µs) compared to the control film (9.70 µs), calculated from tran-
sient photoluminescence spectroscopy (TRPL, excitation at 500 nm) in
Supplementary Fig. 24 and Supplementary Table 3, elucidates the
facilitated electron transfer and reduced defect recombination[37].

## Device performance of SnF₂-free Sn-Pb PSCs

To evaluate the impact of PbF₂ post-treatment on the photovoltaic
performance of PSCs, we fabricate a series of devices. As shown in
Fig. 5a, Sn-Pb PSCs employing effective lead powder as a reductant and
ethanediamine (EDA) post-treatment exhibit remarkably low PCE of
16.43% when devoid of the SnF₂ additive (control)[12,13,30]. This is pri-
marily attributed to the low open-circuit voltage ($V_{OC}$) of 0.736 V and
fill factor (FF) of 69.56%. In contrast, when the SnF₂-free perovskite film
is post-treated with PbF₂ (target), even without EDA post-treatment,
the PCE significantly improves to 24.07% accompanied by a high $V_{OC}$ of
0.884 V and an FF of 81.3% (Supplementary Fig. 25 and Supplementary
Table 4). The large $V_{OC}$ and FF difference is attributed to the misaligned
energy level and large defect recombination loss in the control device

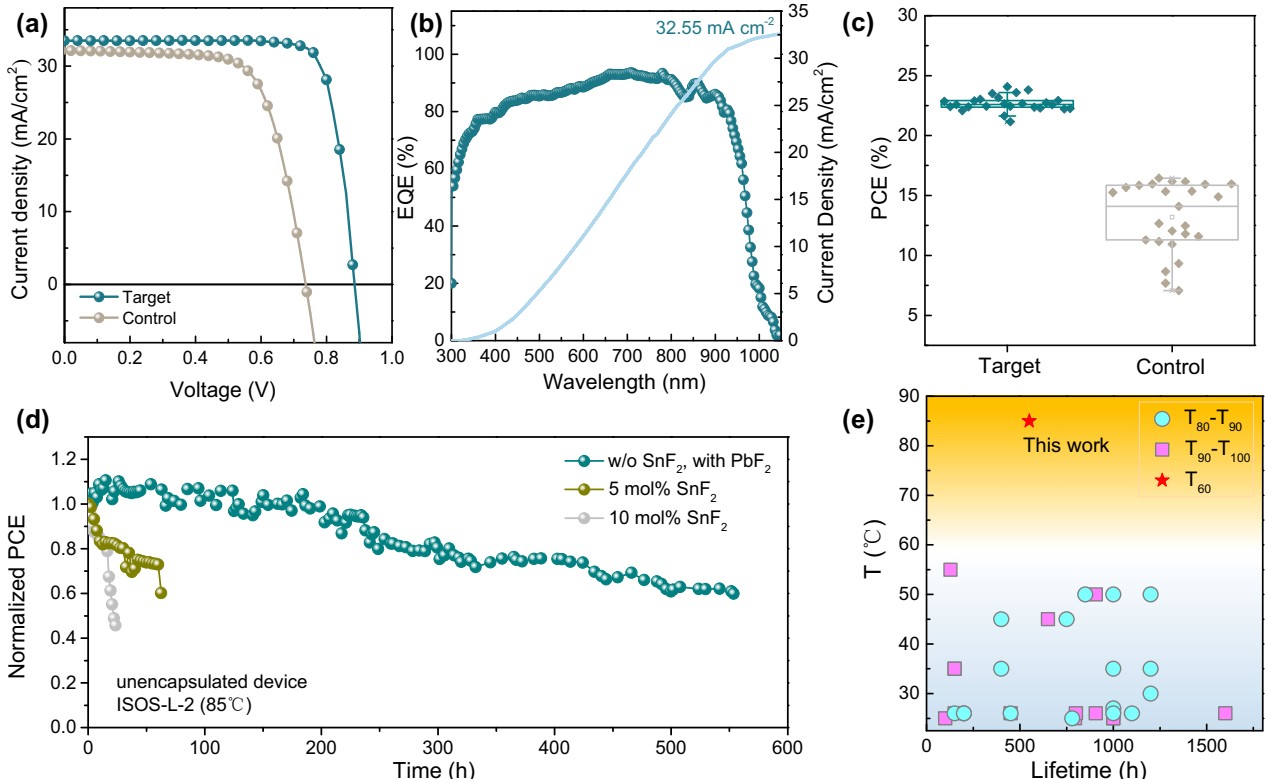

**Fig. 5 | Devices performance. a** *J-V* curves of control and target PSCs. **b** EQE spectra as a function of monochromatic wavelength recorded for target PSC. **c** The statistical PCE of 25 randomly selected target and control devices. **d** The MPP tracing stability of unencapsulated devices in $N_2$ at 85 °C. **e** The reported MPP tracing stability of Sn-Pb PSCs at different temperatures[2,5–7,11,14–17,20,21,23,41–47]. $T_n$ represents the time that device degrades to $n$% of initial PCE.

(Fig. 4c, d). Especially, the $V_{OC}$ shortfall in the control device exceeds its PLQY deficit, a discrepancy chiefly rooted in its misaligned energy levels caused by the p-type defects at film surface[38,39]. The target device demonstrates a steady-state PCE of 23.70% under MPP condition with for 600 s, with minimal hysteresis (Supplementary Fig. 26 and Supplementary Table 5). The external quantum efficiency (EQE) spectrum of the target device, shown in Fig. 5b, exhibits an integrated short-circuit current density ($J_{SC}$) of 32.55 mA cm$^{-2}$, confirming the consistent of the integrated current density from *J-V* characteristic. Figure 5c and Supplementary Fig. 27 present the statistical parameters of 25 control and 25 target devices selected by random sampling. The narrow distributions and small standard deviations underscore the good reproducibility. All target devices exhibit simultaneous gains in $V_{OC}$ and FF, yielding a marked and consistent PCE enhancement after PbF$_2$ post-treatment, which firmly corroborates the reliability of this interfacial strategy. The performance of devices post-treated with other lead salts is also investigated (Supplementary Fig. 28 and Supplementary Table. 6). Devices post-treated by PbCl$_2$ and PbBr$_2$ show only a modest increase in V$_{OC}$ and a minimal improvement in the ultimate PCE. This suggests the crucial role of F$^-$ ions in suppressing Sn surface defects and highlights their indispensability for enhancing device performance. To map the full efficiency ceiling, we also test PbF$_2$ & EDAI$_2$ post-treatment (Supplementary Fig. 29 and Supplementary Table. 7), SnF$_2$ additive + PbF$_2$ post-treatment (Supplementary Fig. 30 and Supplementary Table 8), and PbF$_2$ as an additive (Supplementary Figs. 31, 32 and Supplementary Table 9). It is noteworthy that PbF$_2$ additive lags behind SnF$_2$ because Pb$^{2+}$ binds F$^-$ too tightly, leaving fewer F$^-$ to complex with Sn$^{4+}$ in precursor and block its entry into the perovskite lattice[40]. The primary motivation for replacing the SnF$_2$ additive is to overcome the stability bottleneck of Sn-Pb PSCs, especially under the light-thermal conditions. In this regard, the target device retains 60% of its initial efficiency after continuous operates at 85 °C under MPP

condition for 550 h. In comparison, devices with 10 mol% and 5 mol% SnF$_2$ additives degrade to 60% of their initial efficiency after 19 h and 62 h, respectively (Fig. 5d). Supplementary Fig. 33 tracks the $V_{OC}$, $J_{SC}$, and FF evolution during 85 °C MPP aging. Devices with SnF$_2$ show a sharp FF drop, mirroring the energy-level shift in Fig. 3b. At 10 mol% SnF$_2$, the $V_{OC}$ falls even faster because surface F$^-$ is consumed, leaving V$_{Sn}$ defects. This result aligns with recent studies showing significant improvements in light-soaking and thermal stability, which are correlated with a reduction in SnF$_2$ content by half[41]. This enhancement in stability is particularly noteworthy within the realm of Sn-Pb PSCs. Photo-thermal stability data are rarely documented, with most reported operational temperatures being below 60 °C (Fig. 5e and Supplementary Table 10)[2,5–7,11,14–17,20,21,25,41–47]. Our findings highlight the potential of the combination of PbF$_2$ post-treatment and lead powder in precursor as a highly effective strategy to improve both the efficiency and long-term stability of Sn-Pb PSCs, addressing critical challenges in their practical application.

## Discussion

In summary, we discover that SnF$_2$ additive in Sn-Pb perovskite could accelerate device degradation through reaction between SnF$_2$ and FAI under thermal treatments. Considering the lower reactivity between PbF$_2$ and FAI, lead powder in precursor combining with a PbF$_2$ post-treatment strategy is used to replace the role of SnF$_2$ in the film formation and surface defect passivation, which address the trade-off between efficiency and stability. Employing this strategy, we firstly improved the efficiency of SnF$_2$-free Sn-Pb PSCs from 16.43% to 24.07%, along with improved photo-thermal stability: remaining 60% initial efficiency after operating at 85 °C MPP condition for 550 h. This study identifies the key factors underlying the poor stability of Sn-Pb single-junction and all-perovskite tandem solar cells, potentially offering constructive insights into overcoming their stability bottlenecks.

## Methods

### Materials

All the chemicals in this work were used as received from commercial without further purified. $CH(NH_2)_2I$ (FAI) (99.9%), PC60BM (99.9%) were purchased from Advanced Election Technology CO. Ltd. Guanidinium thiocyanate (GASCN) (≥99.5%), $PbI_2$ (99.99%) C60 (>99%) and BCP (>99%) were acquired from the Xi'an Polymer Light Technology. $SnI_2$ (99.999%), $SnF_2$ (>99%), isopropanol (99.5%), ethylenediamine dihydroiodide ($EDAI_2$) (≥99%) and ethylenediamine (EDA) (≥99%) were purchased from Sigma-Aldrich. $PbF_2$ (99.99%) was purchased from Boer. Poly [3-(4-carboxylbutyl) thiophene] (P3CT) was purchased from Rieke, America. Lead powder (99.95%) and $CsOH·H_2O$ (99.9%) were achieved from Aladdin. All solutions were filtered with 0.22 μm PTFE filter before use.

### Solar cell fabrication

ITO glass substrates were cleaned by detergent, DI water, acetone and isopropanol for 20 min respectively via sonication treatment. Then the substrates were treated by UV-zone for 20 min before the deposition of P3CT-Cs HTL. P3CT-Cs solution was prepared through dissolving 5 mg P3CT and 4.62 mg $CsOH·H_2O$ in methanol. The 0.5 mg/ml P3CT-Cs solution was spin-coated on ITO substrates at 2000 rpm for 30 s. Then, the P3CT-Cs film was annealed at 100 °C in air for 10 min. Then the substrates were transferred to glovebox for the deposition of perovskite film. For the Sn-Pb mixed perovskite precursor with $SnF_2$ additive, 1.2 mmol FAI, 0.3 mmol CsI, 0.75 mmol $SnI_2$, 0.075 mmol $SnF_2$, 0.75 mmol $PbI_2$ and 0.025 mmol GASCN were dissolved in 600 μl DMF and 200 μl DMSO mixed solvent. For the Sn-Pb mixed perovskite precursor without $SnF_2$ additive, 0.075 mmol $SnF_2$ is removed and other precursors remain unchanged. For the Sn-Pb mixed perovskite precursor adding lead powder, a slight excess of FAI is employed, as a portion is consumed in the in-situ reaction with the lead powder. After adding 20 mg lead powder, the precursor solution was stirred at room temperature overnight to prove the sufficient reaction between precursor and lead powder. The Sn-Pb mixed perovskite film was spin-coated on P3CT-Cs layer at 1000 rpm for 10 s and 4000 rpm for 50 s. 300 μl chlorobenzene was in-situ dripped onto the perovskite film after 45 s during the second step within 1 s. Afterwards, the perovskite films were immediately transferred to the hotplate and were annealed at 130 °C for 7 min. The ambient temperature during film formation is controlled at 25 °C to prevent excessively rapid crystallization of the perovskite. To the perovskite films with $SnF_2$ additive, post-treatment with 0.1 mM EDA in chlorobenzene at 5000 rpm for 20 s is conducted. To the perovskite films without $SnF_2$ additive, post-treatment with 1.0 mg/ml $PbF_2$ in isopropanol at 5000 rpm for 20 s is conducted. The post-treatment with $PbCl_2$ and $PbBr_2$ is conducted by evaporating 1 nm film on perovskite. The combined post-treatment with $PbF_2$ and $EDAI_2$ is conducted by a mixed solution of 1.0 mg/ml $PbF_2$ and 0.2 mg/ml $EDAI_2$ in isopropanol at 5000 rpm for 20 s. Then, a further treatment at 100 °C was carried out for 5 min. Finally, 30 nm C60, 6 nm BCP, 1 nm LiF and 100 nm Cu electrode were evaporated under high vacuum ($<3*10^{-4}$ Torr). The device area defined by the mask was 0.0836 $cm^2$.

### Characterization and measurements

Photocurrent density-voltage (J-V) curves (Keithley 2400 Source-Meter) were measured under one sun illumination (AM1.5 G, 100 mW/$cm^2$) using the solar simulator (Enlitech, SS-F5-3A). The light source is a 450-W xenon lamp calibrated by a standard Si reference solar cell (Enli/SRC2020, SRC-00201). The J-V curves were measured from 1.0 V to −0.2 V with dwell time of 10 ms. The light source is a 45-W xenon lamp calibrated by a standard Si reference solar cell (Enli/SRC2020, SRC-00201). The EQE measurement was carried out in ambient air using a QE system (SOFN Instruments Co., Ltd) with monochromatic light focused on a device pixel. The MPP stability test was conducted on encapsulated devices in $N_2$ under 1-sun equivalent illumination (white light-emitting diodes) at 85 °C which was monitored by an infrared thermometer.

TOF-SIMS spectra was measured on aged samples using PHI nanoTOF II Time-of-Flight SIMS with a primary Bi ion gun (30 keV). 1 keV Cs gun was used for sputtering with an analysis area of $200 × 200$ $μm^2$. The aged perovskite devices were prepared by operating at MPP at 85 °C under 1-sun equivalent illumination (white light-emitting diode) in $N_2$.

X-ray photoelectron spectroscopy (XPS) and ultraviolet photoelectron spectroscopy (UPS) measurements were carried out using Kratos AXIS ULTRA DALD XPS/UPS system. XRD spectra was recorded in an Empyrean Micro diffractometer with Cu Kα radiation. Scanning electron microscope (SEM) images were acquired by using a field-emission scanning electron Microscopy (Zeiss GeminiSEM450).

TGA-MS is conducted using a Netzsch simultaneous thermal analysis (STA) instrument, under a nitrogen atmosphere created by fluxing 50 mL $min^{-1}$ of $N_2$ at the heating rate of 10 °C $min^{-1}$. TG-MS was conducted using a Netzsch simultaneous thermal analysis instrument coupled with a gas chromatography-mass spectrometer where the mass of any volatile fragments can be analyzed and identified. The powder samples were heated from 25 °C to 300 °C with a ramp rate of 10 °C/min.

Activation energy measurement of ions migration: the current was extracted at 100 s after the voltage is switched on. The measurement was conducted in a Lakeshore Probe Station under vacuum ($4.0 × 10^{-4}$ Pa). The samples were placed on a copper substrate with temperature control by a heater and injected liquid He. A semiconductor characterization system (Malaysia TEK BA1500) was used for the current measurement. During the measurement, we first cooled the devices to 165 K for 1 h and then heated to objective temperature. Every objective temperature was stabilized for 5 min before the current record.

The steady-state and time-resolved PL spectra were measured utilizing FLS980E Fluorescence Spectrophotometer (excited by 532 nm, Unite Kingdom) at FOMAD (Family of Master and Doctor) Corp, China. The PLQY was performed using an Edinburgh Instruments FS5 spectrofluorimeter using a plane-ruled diffraction grating monochromators with 1200 grooves/mm coupled to filter turrets to remove higher order diffraction signals and using a Hamamatsu R928P photomultiplier tube as a primary detector and two silicon photodiode detectors for reference and transmission detection. A Spectralon integrating sphere attachment was used for PLQY measurements with an excitation wavelength of 520 nm and excitation fluence of 1 mW/$cm^2$. For fs-TAS measurements, the Titanium:Sapphire femtosecond laser (Astrella, Coherent Inc) generated femtosecond pulsed light with a central wavelength of 800 nm, a repetition rate of 1 kHz, and a pulse width of 100 fs. This 800 nm pulsed light was split into two parts by a beam splitter. One part entered the optical parametric amplifier (OPerA Solo, Coherent Inc) to produce the pump light at 600 nm (70 W). The other part passed through a delay line (Delayline) and entered the transient absorption spectrometer (Helios, Ultrafast system), where it was focused on a sapphire nonlinear to generate a supercontinuum probe light. The probe light, after being focused by an off-axis parabolic mirror, was directed onto the sample. The pump light, after being chopped to 500 Hz by a chopper, was adjusted by a half-wave plate to ensure that the pump and probe lights were focused onto the sample at an angle of 54.7°. During testing, the sample was clamped on a sample stage that can move two-dimensionally in the direction perpendicular to the beam to reduce damage to the sample from the pump light). The probe light absorbed by the sample carried information about the particle's excited state and ground state and was ultimately incident on the optical fiber probe head. By using a time delay line to change the delay of the probe light relative to the pump light reaching the sample, spectral information of the sample particles at different time delays after excitation can be obtained.

## Reporting summary

Further information on research design is available in the Nature Portfolio Reporting Summary linked to this article.

## Data availability

The data generated in this study are provided in the Supplementary Information/Source Data file and have been deposited in the Figshare repository (https://doi.org/10.6084/m9.figshare.29987002)[48]. Source data are provided with this paper.

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

## Acknowledgements

This work was supported by National Science Fund for Distinguished Young Scholar (No. T2325011, J.F.), National Natural Science Foundation of China (No. 62374058, J.F., No. 52173161, J.F., No. 62274062, X.L.), Shanghai Science and Technology Innovation Action Plan (No. 24DZ3001202, X.L.), the Fundamental Research Funds for the Central Universities and National Youth Top-notch Talent Support Program, Materials Characterization Center, ECNU Multifunctional Platform for Innovation.

## Author contributions

These authors contributed equally: Haobo Yuan, Wenxiao Zhang. J.F and F.G. supervised the whole project. H.Y. and W.Z. conceived the idea and designed and participated in all the experimental sections. J.X., X.G., Y.L. and B.F. help to conduct the stability, TG and SEM measurement. Y.H., Z.C. and W.L. help to conduct and analysis the XPS and UPS measurement, S.F. and X.L. help to analyze the results. H.Y. and W.Z. co-wrote original draft. J.F. and F.W. reviewed and edited the final manuscript.

## Competing interests

The authors declare no competing interests.
