## [Transparent Peer Review file · Nature Communications]

A Tin Fluoride-Free, Efficient and Durable Tin-Lead Perovskite Solar Cell

Corresponding Author: Professor Junfeng Fang

Version 0:

Reviewer comments:

Reviewer #1

(Remarks to the Author)

The manuscript by Yuan et. al. reveals the reaction between SnF₂ and FAI in tin-lead perovskite solar cells (Sn-Pb PSCs) during photo-thermal treatments. This degradation reaction reveals the origin of the low stability level of high-efficiency Sn-Pb PSCs and all-perovskite tandem solar cells. However, SnF₂ is an indispensable additive in Sn-Pb PSCs without substitutes. Based on this, authors proposed the novel strategy that combines lead powder in precursor with PbF₂ post-treatment. As a result, the PCE of SnF₂-free Sn-Pb PSCs was greatly improved from 16.43% to 24.07%. More importantly, the photo-thermal stability of device is obtained, remaining at 60% initial efficiency after operating at 85°C at MPP condition for 550 h. The substance and conclusions of this work are likely to generate significant interest within the field. I recommend that Nature Communications accept this manuscript after addressing the following concerns.

1. The authors focus on Sn-Pb perovskite devices without SnF₂ additive and describe the irreplaceability of SnF₂ additive. Have analogous devices fabrication been reported in other studies? Please provide the research status of SnF₂-free devices
2. Will the PbF₂ post-treatment affect the Sn/Pb ratio on the film surface, and what potential impacts might it have on the device performance? Further evidence is needed to clarify this issue.
3. In Figure 1, the authors depicted the SEM morphology of Sn-Pb perovskite films subjected to 85°C light soaking, both with and without SnF₂. However, the morphology characteristics of the perovskite films treated with PbF₂ post-treatment were not addressed. Could you please provide the corresponding information regarding the morphology of the PbF₂ post-treated perovskite films.
4. In Figure 4, the statistical parameters of the perovskite solar cells only include the efficiency. The main text and supplementary information do not provide statistical data for other key performance parameters such as Voc, Jsc, and FF. Please provide this data.
5. The manuscript utilized PLQY to characterize the non-radiative recombination loss on the film surface, but the increase in PLQY does not be consistent with the increase in the Voc of device. Please explain that.

Reviewer #2

(Remarks to the Author)

The poor photo-thermal stability of Sn-Pb perovskite solar cells, especially at 85°C, is dragging back the further development of Sn-Pb single-junction devices and all-perovskite tandem devices. Impressively, Yuan et al. discovered the thermal degradation process in Sn-Pb PSCs induced by SnF₂. Further, the reductive and crystallization regulation effect of lead powder in combination with the energy level regulation effect of PbF₂ ensured the device performance and thermal stability, simultaneously. Under continuous operation at the maximum power point (MPP) at 85°C, the devices maintained 60% of their initial efficiency after 550 hours. The manuscript discovered an important phenomenon and provided a promising solution to resolve the instability issue of Sn-Pb perovskite. Therefore, I suggest that Nature Communications accept this manuscript after minor revision.

1. As a widely utilized antioxidant, SnF₂ is capable of passivating Sn⁴⁺ via a substitution reaction. Can PbF₂ possess a comparable function?
2. The post-treatment of PbF₂ improved the surface work function (Supplementary Figure 12), however, the uniformity of this modification has not been proven. Please provide the surface potential mapping of KPFM.
3. Did the authors use classic surface passivators such as EDADl2 when conducting device post-processing with PbF₂?

Can the device performance be further improved when PbF₂ and EDADI2 act together?

4. In addition to the efficiency evolution, it is also necessary to provide the evolution of other performance parameters (including Voc, Jsc, and FF) during operation at the maximum power point (MPP) and 85°C. Furthermore, an interpretation of the evolution process would be highly appreciated.

5. The performance degradation pathways of Sn-Pb PSCs are diverse. How do the authors consider the stability potential of Sn-containing PSCs at 85°C?

Reviewer #3

(Remarks to the Author)

This work demonstrates an impressive PCE exceeding 24% for SnF₂-free tin-lead perovskite solar cells, and importantly, investigates a critical degradation pathway induced by SnF₂ that has been previously overlooked. To address this issue, the authors propose a promising strategy involving the use of PbF₂, which significantly enhances both the open-circuit voltage and fill factor of the devices. Notably, the devices also exhibit improved thermal stability at 85 °C under maximum power point tracking, even without encapsulation, underscoring their potential for practical applications. However, the manuscript requires revision, particularly in terms of language clarity and coherence. In addition, the experimental procedures should be described in greater detail to provide a comprehensive understanding of the methodology employed. Regarding the scientific and technical significance of the work, I support its publication in Nature Communications after the authors address the following points and provide the necessary additional information:

1. Tin-lead perovskites are highly sensitive to the crystallization environment, and the absence of SnF₂ can significantly influence the film formation process. Therefore, a detailed description of the device fabrication protocol is essential.
2. Did the authors investigate the effect of varying PbF₂ concentrations? If so, please include the relevant data and discuss the optimal concentration range.
3. The manuscript lacks information regarding the morphology of the perovskite films after PbF₂ post-treatment, both in their fresh and aged states. Please provide SEM or AFM images and related analysis.
4. The authors claim that the by-product HF reacts with the ITO electrode. However, there is currently insufficient evidence to substantiate this. Please provide experimental data to support this assertion.
5. While the experimental results clearly show the superiority of PbF₂ in improving the performance of SnF₂-free Sn-Pb PSCs, a more systematic explanation is needed. The authors should provide further clarification through theoretical analysis or defect characterization techniques.
6. Can PbF₂ be used as a dopant in the precursor solution instead of SnF₂? If so, how does this affect device performance? Please include a comparative study.
7. The use of Pb powder in the precursor solution, rather than Sn powder, is unusual and may raise questions for the reader. Please clarify the rationale behind this choice.

Reviewer #4

(Remarks to the Author)

The manuscript entitled "A Tin Fluoride-Free, Efficient and Durable Tin-Lead Perovskite Solar Cell" reveals that the instability of SnF₂ in Sn-Pb perovskite solar cells originates from its thermal reaction with FAI, which accelerates device degradation. The authors propose an innovative strategy by introducing lead powder in the precursor solution combined with a PbF₂ post-treatment to replace the role of SnF₂ in film formation and surface defect passivation. This approach effectively balances efficiency and stability. As a result, the power conversion efficiency of SnF₂-free devices is improved from 16.43% to 24.07%, with excellent photo-thermal stability—maintaining 60% of the initial efficiency after 550 hours of operation at 85°C under MPP conditions. The study is well-structured, the data are comprehensive, and the findings are of significant importance for the development of both single-junction and all-perovskite tandem solar cells. However, several issues should be thoroughly addressed and resolved prior to publication.

1. The authors identified that SnF₂, when used as an additive, presents issues related to thermal stability, and they accordingly employed metallic Pb to suppress oxidation and used PbF₂ post-treatment for defect passivation. However, the current evidence supporting the instability caused by SnF₂ relies mainly on TG analysis. To strengthen the conclusion, the authors are encouraged to provide further theoretical explanations and complementary characterizations to support this claim. Additionally, it would be valuable to discuss whether similar findings have been reported in the literature.
2. It is recommended that the authors include additional performance data for devices fabricated using PbF₂ as an additive, in order to better justify the rationale for employing PbF₂ as a post-treatment rather than incorporating it directly into the precursor. Moreover, while the authors emphasize the synergistic effect of Pb and PbF₂, data showing the individual contributions of each component are lacking. Further clarification is needed regarding the combined effect of SnF₂ and PbF₂, and whether their interaction is beneficial or detrimental to device performance.
3. Figure 1c reveals pronounced pinholes/cracks in the perovskite film attributed to SnF₂ incorporation. Please provide a mechanistic explanation for this morphological degradation, addressing how SnF₂ influences crystallization kinetics or induces lattice strain.

4、 The textual annotations within the device architecture diagram are unclear in scheme 1 and figure 3a. It is recommended to use more distinct and contrasting labels or visual indicators to clearly differentiate the various layers and components of the solar cell structure. In figure S7 and S12, the “Aged 200 h” curves also should be clarified.

5、 On page 4, the authors report a 25.1% weight loss for the FAI + SnF₂ powders around 500 °C; however, this value is inconsistent with the 25.8% shown in Figure 1a. The authors should clarify this discrepancy to ensure the accuracy and consistency of the data presented.

6、 On page 9, Figure 3d should clearly indicate the conditions corresponding to each colored column.

7、 On page 10, the authors state, “The external quantum efficiency (EQE) spectrum of the target device, shown in Figure 4c, exhibits an integrated short-circuit current density (JSC) of 32.55 mA cm⁻², confirming the consistency of the integrated current density from J-V characteristic.” However, this statement contains two errors: the EQE data are actually shown in Figure 4b, not 4c, and Figure 4c is not referenced or described elsewhere in the manuscript. The authors should revise this sentence to correct the figure number and ensure consistency with the actual content.

Version 1:

Reviewer comments:

Reviewer #1

(Remarks to the Author)

The authors have fully addressed my comments. I support the publication of the revised manuscript.

Reviewer #2

(Remarks to the Author)

The authors have addressed my concerns. I recommend the manuscript for publication.

Reviewer #3

(Remarks to the Author)

The authors have addressed my concerns and made good revisions. I am pleased to recommend publishing this work in Nature Communications in its current form.

Reviewer #4

(Remarks to the Author)

The authors have provided a comprehensive and meticulous response to all points raised by the reviewers. The revisions made to the original manuscript are thorough and appropriate. All key changes have been clearly outlined in the point-by-point response letter and are accurately highlighted within the revised manuscript.

The quality of the manuscript has been significantly enhanced through this revision process. All major and minor concerns previously identified have been addressed satisfactorily. The manuscript now meets the required standards for publication in terms of scientific soundness, clarity, and completeness.

Reviewers' Comments:

Reviewer #1:

The manuscript by Yuan et. al. reveals the reaction between SnF₂ and FAI in tin-lead perovskite solar cells (Sn-Pb PSCs) during photo-thermal treatments. This degradation reaction reveals the origin of the low stability level of high-efficiency Sn-Pb PSCs and all-perovskite tandem solar cells. However, SnF₂ is an indispensable additive in Sn-Pb PSCs without substitutes. Based on this, authors proposed the novel strategy that combines lead powder in precursor with PbF₂ post-treatment. As a result, the PCE of SnF₂-free Sn-Pb PSCs was greatly improved from 16.43% to 24.07%. More importantly, the photo-thermal stability of device is obtained, remaining at 60% initial efficiency after operating at 85°C at MPP condition for 550 h. The substance and conclusions of this work are likely to generate significant interest within the field. I recommend that *Nature Communications* accept this manuscript after addressing the following concerns.

Response:

We would also like to thank the reviewer for the valuable comments and helpful suggestions. In the following pages, we provide point-by-point responses to all the questions raised by the reviewers. The corrections in the manuscript are shown in yellow for convenience. We also carefully read the entire manuscript and revised any errors we found. We hope that we have appropriately addressed all the mentioned issues.

1. The authors focus on Sn-Pb perovskite devices without SnF₂ additive and describe the irreplaceability of SnF₂ additive. Have analogous devices fabrication been reported in other studies? Please provide the research status of SnF₂-free devices.

Response:

We thank the reviewer for this suggestion. SnF₂ is the irreplaceable additive in the high-efficiency Sn-based, Sn-Pb perovskite solar cells as well as the all-perovskite solar cells. In the initial exploration phase of Sn-Pb perovskites, the SnF₂-free devices only delivered efficiencies lower than 10% (*J. Am. Chem. Soc.* **2014**, *136*, 8094-8099; *J. Phys. Chem. Lett.* **2014**, *5*, 1004-1011). At the same time, SnF₂ was doped in the pure

Sn-based precursor to suppress the formation of Sn⁴⁺ defects in the solution-processed film. Introducing 20 mol% SnF₂ in CsSnI₃- and FASnI₃-based films improves the device efficiency from 3*10⁻⁴% to 2.02%, and from 0.02% to 2.10%, respectively. These results indicate the necessity of SnF₂ to tap into the potential for high efficiency. Therefore, the subsequent Sn-Pb perovskite solar cells have also incorporated approximately 10 mol% of SnF₂. Although it has been found that excessive aggregation of SnF₂ negatively impacts device efficiency, no effective methods for completely replacing SnF₂ have been reported. The typical device performance of SnF₂-free Sn-Pb perovskite solar cells is lower than 15% from the recent reported articles (*ACS Energy Lett.* **2019**, *4*, 2748-2756; *Adv. Energy Mater.* **2021**, *11*, 2101045; *J. Phys. Chem. C* **2021**, *125*, 12560-12567; *ACS Appl. Mater. Interfaces* **2023**, *15*, 10150-10157), which is in accordance with the control device in our manuscript. This current status of SnF₂-free devices has now been added to paragraph 2 of the manuscript.

2. Will the PbF₂ post-treatment affect the Sn/Pb ratio on the film surface, and what potential impacts might it have on the device performance? Further evidence is needed to clarify this issue.

Response:

We thank the reviewer for this insightful comment. PbF₂ post-treatment effectively lowers the surface Sn/Pb ratio from 1.17 to 0.78 determined by the SEM with energy-dispersive X-ray spectroscopy (SEM-EDS, Supplementary Fig. S14 and Supplementary Tab. 1). Replacing Sn²⁺ with Pb²⁺ suppresses tin vacancies (V_{Sn}) formation because the deeper 5s² states of Pb²⁺ yield stronger Pb-I bonds that resist cleavage, whereas Sn-I bonds are readily broken (*Adv. Mater.* **2019**, *31*, 1803792.).

Supplementary Table 1. The Pb and Sn atomic concentration and Sn/Pb atomic ratio obtained in SEM-EDS mapping.

Sample	Pb (At %)	Sn (At %)	Sn/Pb
Pristine	5.7	6.7	1.17
PbF ₂ Post-treatment	5.5	4.4	0.78

Supplementary Figure 14. The Energy Dispersive X-Ray Spectroscopy maps of perovskite films: (a)pristine and (b) after PbF₂ post-treatment.

In addition, PbF₂ remains chemically inert toward FAI during both fabrication and operation, suppressing A-site vacancies (V_{FA}). Both V_{FA} and V_{Sn} defects are p-type defects that can pin the Fermi level (E_F) near the valence band (VB) edge: V_{FA} introduces shallow, delocalized states, while V_{Sn} generates deep, localized states that intensify non-radiative recombination. High concentration of V_{Sn} create a locally iodine-rich environment, while V_{FA} lowers the I⁻ ion migration barrier by reducing steric hindrance, thereby facilitating the formation of further iodine-related defects p-

type defects (such as I_{Sn} and I_{FA}). Consequently, these surface p-doping defects are mutually reinforcing (*ACS Energy Lett.* **2024**, *9*, 400-409). Their accumulation bends the bands upward at the perovskite/C60 interface, degrading the quasi-Fermi-level splitting (QFLS). Their accumulation bends the bands upward at the perovskite/C60 interface. By mitigating such p-type defects, PbF_2 post-treatment forms a matching and stable energy level structure at the perovskite/C60 interface (Supplementary Fig. 8 and 20). We also supplement this interpretation on page 8 and 9 in the manuscript.

Supplementary Figure 8. (c) UPS spectra of Sn-Pb perovskite without SnF_2 additive before and after aging at 85°C with light soaking for 200 h.

Supplementary Figure 20. The UPS spectra of the perovskite with (a) SnF₂ post-treatment and (b) PbF₂ post-treatment.

3. In Figure 1, the authors depicted the SEM morphology of Sn-Pb perovskite films subjected to 85°C light soaking, both with and without SnF₂. However, the morphology characteristics of the perovskite films treated with PbF₂ post-treatment were not addressed. Could you please provide the corresponding information regarding the morphology of the PbF₂ post-treated perovskite films.

Response:

We thank the reviewer for this valuable comment. We have included SEM images of Sn-Pb perovskite films without SnF₂ additive and with PbF₂ post-treatment in the Supplementary Materials (Supplementary Fig. 18) to illustrate their morphology before and after aging under 85 °C light soaking conditions. The fresh PbF₂ post-treated perovskite films exhibit a morphology that closely resembles that of the fresh perovskite films without SnF₂ additive, as shown in Supplementary Fig. 3. Upon aging under 85 °C light soaking, the morphology of the PbF₂ post-treated perovskite films remains nearly unchanged, mirroring the aged perovskite films without SnF₂ additive depicted in Fig. 1d. These observations suggest that the lack of chemical reaction between PbF₂ and the perovskite components effectively prevents the deterioration of the perovskite morphology.

Supplementary Figure 18. SEM spectra of Sn-Pb perovskite film without SnF₂ additive and with PbF₂ post-treatment before and after aging at 85 °C with light soaking for 200 h.

4. In Figure 4, the statistical parameters of the perovskite solar cells only include the efficiency. The main text and supplementary information do not provide statistical data for other key performance parameters such as V_{oc} , J_{sc} , and FF. Please provide this data.

Response:

We thank the reviewer for this suggestion. We have supplemented the statistical data for parameters including V_{oc} , J_{sc} , and FF of 25 randomly selected target and control perovskite solar cells (Supplementary Fig. S27) corresponding with the statistical PCE in Fig. 4c.

Supplementary Figure 27. The statistical (a) V_{oc} , (b) J_{sc} and (c) FF of 25 randomly selected target and control devices.

5. The manuscript utilized PLQY to characterize the non-radiative recombination loss on the film surface, but the increase in PLQY does not be consistent with the increase in the V_{oc} of device. Please explain that.

Response:

We thank the reviewer for this valuable comment. The PLQY can be employed to calculate the quasi-Fermi levels splitting (QFLS) in perovskite solar cells, which is useful metric for evaluating defect recombination losses at interfaces within devices. However, a mismatch between QFLS and open-circuit voltage (V_{oc}) is often observed, particularly in devices with poor performance. In high-efficiency cells, however, V_{oc} typically equals QFLS. One primary reason for this mismatch is the energy level misalignment at the interfaces, which creates a transfer barrier for the majority carriers and increases the minority carrier population at the interface. This, in turn, leads to an exponential rise in the recombination rates and a direct loss in V_{oc} (*Adv. Energy Mater.*

2023, 13, 2303135; *Adv. Energy Mater.* 2019, 9, 1901631; *Energy Environ. Sci.*, 2019, 12, 2778-2788).

In this study, perovskite film without SnF₂ additive exhibit a downshift of the Fermi level downshift at the surface (Supplementary Fig. 7 and 8c-d), which is attributed to the presence of V_{Sn}, V_{FA} p-type defects. This energy level misalignment increase the electron transfer barrier and lead to hole accumulation, thereby enhancing interfacial recombination losses and resulting in a significant QFLS/*V*_{oc} mismatch. Conversely, in perovskite films with SnF₂ additive and PbF₂ post-treatment, the p-type defects are significantly suppressed, and the energy level alignment at the perovskite/C60 interfaces is improved. This facilitates electron transfer and effectively repels holes away from this interface region. In these low-surface-recombination solar cells, the QFLS/*V*_{oc} mismatch is minimized. As a result, the increased in PLQY for devices with SnF₂ additive and PbF₂ post-treatment is not directly correlated with the increase in *V*_{oc}. In other words, PLQY primarily reflects differences in defect recombination losses and does not account for recombination losses caused by energy level misalignment. Therefore, in this manuscript, PLQY is used solely to compare defect recombination losses at different perovskite/C60 interfaces and should not be employed to compare *V*_{oc} differences among various devices. We also supplement this analyses to clarify the *V*_{oc} differences among different devices in Figure 4a in the manuscript.

Reviewer #2:

The poor photo-thermal stability of Sn-Pb perovskite solar cells, especially at 85°C, is dragging back the further development of Sn-Pb single-junction devices and all-perovskite tandem devices. Impressively, Yuan *et al.* discovered the thermal degradation process in Sn-Pb PSCs induced by SnF₂. Further, the reductive and crystallization regulation effect of lead powder in combination with the energy level regulation effect of PbF₂ ensured the device performance and thermal stability, simultaneously. Under continuous operation at the maximum power point (MPP) at 85°C, the devices maintained 60% of their initial efficiency after 550 hours. The manuscript discovered an important phenomenon and provided a promising solution to resolve the instability issue of Sn-Pb perovskite. Therefore, I suggest that *Nature Communications* accept this manuscript after minor revision.

Response:

We would also like to thank the reviewer for the valuable comments and helpful suggestions. In the following pages, we provide point-by-point responses to all the questions raised by the reviewers. The corrections in the manuscript are shown in yellow for convenience. We also carefully read the entire manuscript and revised any errors we found. We hope that we have appropriately addressed all the mentioned issues.

1. As a widely utilized antioxidant, SnF₂ is capable of passivating Sn⁴⁺ via a substitution reaction. Can PbF₂ possess a comparable function?

Response:

We thank the reviewer for this valuable comment. The resistance of oxidation of SnF₂ and PbF₂ when used as additives in the tin-lead perovskite precursor solution is compared. As show in the orange color and UV-vis absorption spectra of the pristine SnI₂ solution, indicating the presence of SnI₄ (Supplementary Fig. 31). Upon introducing 10 mol% SnF₂ (based on the molar ratio to SnI₂), the solution changes from orange to yellow. This color change occurs because F⁻ has a stronger affinity of for Sn⁴⁺ than Sn²⁺, leading to the selective formation of SnF₄ complexes (*Angew. Chem. Int. Ed.* **2021**, *60*, 21583). SnF₄ has a lower tendency than SnI₄ to incorporate into the perovskite

structure, thereby preventing the formation of tin vacancy defects and improving device performance. In contrast, when PbF_2 is introduced, the strong bonding between Pb^{2+} and F^- makes it difficult for F^- to complex with Sn^{4+} . As a result, PbF_2 is less effective than SnF_2 in preventing Sn^{4+} from being incorporated into the perovskite structure. Consequently, after adding into 10 mol% PbF_2 (based on the molar ratio to SnI_2), the color of the solution remains almost unchanged (Supplementary Fig. 31). The device performance with PbF_2 as an additive is also less competitive as that with SnF_2 additive, as demonstrated in Supplementary Fig. 32 and Supplementary Tab. 9. We also supplemented this data and discussion in the manuscript and Supplementary Materials.

Supplementary Figure 31. The UV-vis absorption spectra of tin-lead perovskite precursor solution with SnF_2 (a) and PbF_2 (b) additive aging in air. (c) The color changes of SnI_2 with SnF_2 and PbF_2 additive.

Supplementary Figure 32. *J-V* curves of Sn PSCs with different concentration PbF_2 additive.

Supplementary Table 9. Summary of photovoltaic performance of Sn-Pb perovskite PSCs with different concentration PbF_2 additive.

PbF_2 conc. (mol% vs. perovskite)	V_{oc} (V)	J_{sc} (mA/cm^2)	FF (%)	PCE (%)
0	0.73	32.1	69.5	16.4
2.5	0.866	31.1	62.9	17.02
5.0	0.885	32.6	66.1	19.1
7.5	0.885	34.6	51.7	15.08

2. The post-treatment of PbF_2 improved the surface work function (Supplementary Figure 12), however, the uniformity of this modification has not been proven. Please provide the surface potential mapping of KPFM.

Response:

We thank the reviewer for this valuable comment. In response to the reviewer's suggestion, we employ the KPFM to investigate the potential change of the perovskite film surface before and after PbF_2 post-treatment. The PbF_2 post-treatment caused the surface potential of the perovskite to increase from -195.2 eV to -135.7 eV. This demonstrates that PbF_2 post-treatment can passivate the harmful p-type defects at the perovskite surface. At the same time, the PbF_2 post-treatment makes the interface potential more uniform, indicating less carrier accumulation. The uniform surface potential is beneficial for further improving device performance.

Supplementary Figure 22. The Atomic Force Microscopy (AFM) and Kelvin Probe Force Microscopy of Sn-Pb perovskite film with (a-b) and without (c-d) PbF_2 post-treatment.

3. Did the authors use classic surface passivates such as EDAI₂ when conducting device post-processing with PbF₂? Can the device performance be further improved when PbF₂ and EDAI₂ act together?

Response:

We thank the reviewer for this valuable comment. As a commonly utilized interfacial modifier, ethylenediamine dihydroiodide (EDAI₂) is broadly applied to improve the open-circuit voltage (V_{oc}) of Sn-Pb PSCs. The following figure illustrates the device performance after co-treatment with PbF₂ and EDAI₂. While the incorporation of EDAI₂ results in a slight increase in V_{oc} by 0.04 V, excessive passivation adversely affects the fill factor (FF), leading to a notable reduction. Consequently, the power conversion efficiency (PCE) of the device treated with both PbF₂ and EDAI₂ is 22.5%, which is lower than that of the device treated solely with PbF₂ (24.07%).

Supplementary Figure 29. The J - V curves of Sn-Pb PSCs with post-treatment and post-treatment in combination of PbF₂ and EDAI₂.

Supplementary Table 7. Photovoltaic performance of Sn-Pb perovskite PSCs with post-treatment in combination of PbI₂ and EDAI₂.

post-treatment	V_{oc} (V)	J_{sc} (mA/cm ²)	FF (%)	PCE (%)
PbF ₂ /EDAI ₂	0.888	33.48	75.6	22.5
PbF ₂	0.884	33.5	81.3	24.07

4. In addition to the efficiency evolution, it is also necessary to provide the evolution of other performance parameters (including V_{oc} , J_{sc} , and FF) during operation at the maximum power point (MPP) and 85°C. Furthermore, an interpretation of the evolution process would be highly appreciated.

Response:

We thank the reviewer for this suggestion. We have supplemented the evolution of other performance parameters, including V_{oc} , J_{sc} , and FF, during MPP operation at MPP and 85 °C in the Supplementary Materials (Supplementary Fig. 33). Devices with SnF₂ exhibit a pronounced FF drop upon aging at 85 °C under MPP, consistent with the concomitant energy-level misalignment (Fig. 2b). At 10 mol% SnF₂ (relative to SnI₂), the V_{oc} declines even faster because reaction consumption of F⁻ at the surface and the resulted tin vacancies defects. This mechanistic interpretation of the parameter evolution has now been incorporated into the manuscript.

Supplementary Figure 33. The evolution of V_{oc} , J_{sc} , and FF during MPP tracing of unencapsulated devices in N_2 at 85 °C.

5. The performance degradation pathways of Sn-Pb PSCs are diverse. How do the authors consider the stability potential of Sn-containing PSCs at 85°C?

Response:

We thank the reviewer for this comment. The performance degradation pathways of Sn-Pb PSCs under thermal stress are multifaceted. These pathways include the external environmental oxidation of Sn^{2+} , the formation of tin vacancy defects, the volatilization of A-site organic cations (such as MA^+), and the reaction between the perovskite and acid transport layers (such as PEDOT : PSS). Among these issues, the volatility of A-site organic cations and the erosive nature of transport layers are

common challenges also faced by Pb-based PSCs. The oxidation of Sn^{2+} can be mitigated through appropriate encapsulation techniques. The tin vacancy defects, which typically reside at the film surface, can be suppressed by effective surface passivation methods, such as the use of diamines, oxyacids, 2D perovskites, or Pb-based perovskite heterojunctions. However, the low formation energy of tin vacancy defects promotes the formation of Sn^{4+} and its redox reaction with I_2 , even in an inert environment. This internal cyclic degradation reaction is the key driver behind the rapid degradation of Sn-Pb perovskite films. Strategies such as trapping iodide ions and employing reducing agents hold great potential and are worthy of in-depth investigation.

At the current stage of research, Sn-Pb perovskite films are particularly unstable under thermal stress, leading many researchers to conclude that the degradation is too severe to be resolved. However, in this manuscript, we uncover an overlooked degradation reaction between FAI and the indispensable SnF_2 additive under thermal stress. This reaction consumes perovskite components and generates corrosive by-products that decompose the perovskite structure, which is the root cause of the extreme instability. Once this issue is addressed, the stability challenges of Sn-Pb perovskite films become comparable to those of Pb-based perovskites, indicating significant potential for improvement through more systematic studies.

Reviewer #3:

This work demonstrates an impressive PCE exceeding 24% for SnF₂-free tin-lead perovskite solar cells, and importantly, investigates a critical degradation pathway induced by SnF₂ that has been previously overlooked. To address this issue, the authors propose a promising strategy involving the use of PbF₂, which significantly enhances both the open-circuit voltage and fill factor of the devices. Notably, the devices also exhibit improved thermal stability at 85 °C under maximum power point tracking, even without encapsulation, underscoring their potential for practical applications. However, the manuscript requires revision, particularly in terms of language clarity and coherence. In addition, the experimental procedures should be described in greater detail to provide a comprehensive understanding of the methodology employed. Regarding the scientific and technical significance of the work, I support its publication in *Nature Communications* after the authors address the following points and provide the necessary additional information:

Response:

We would also like to thank the reviewer for the valuable comments and helpful suggestions. In the following pages, we provide point-by-point responses to all the questions raised by the reviewers. The corrections in the manuscript are shown in yellow for convenience. We also carefully read the entire manuscript and revised any errors we found. We hope that we have appropriately addressed all the mentioned issues.

1. Tin-lead perovskites are highly sensitive to the crystallization environment, and the absence of SnF₂ can significantly influence the film formation process. Therefore, a detailed description of the device fabrication protocol is essential.

Response:

We thank the reviewer for this valuable suggestion. The key factors for controlling the crystallization of SnF₂-free tin-lead perovskite films include the amount of lead powder and its reaction time with the perovskite precursor, a slight excess of FAI for its consumption when reacting with lead powder, the ambient temperature during film formation, the volume and dripping duration of the anti-solvent, and the time interval

between anti-solvent dripping and annealing. Here, we provide a detailed description of the device preparation process, with a particular focus on the film formation process of the SnF₂-free tin-lead perovskite, as follows:

For the Sn-Pb mixed perovskite precursor with SnF₂ additive, 1.2 mmol FAI, 0.3 mmol CsI, 0.75 mmol SnI₂, 0.075 mmol SnF₂, 0.75 mmol PbI₂ and 0.025 mmol GASCN were dissolved in 600 μ l DMF and 200 μ l DMSO mixed solvent. For the Sn-Pb mixed perovskite precursor without SnF₂ additive, 0.075 mmol SnF₂ is removed and other precursors remain unchanged. For the Sn-Pb mixed perovskite precursor adding lead powder, a slight excess of FAI is employed, as a portion is consumed in the in-situ reaction with the lead powder. After adding 20 mg lead powder, the precursor solution was stirred at room temperature overnight to prove the sufficient reaction between precursor and lead powder. The Sn-Pb mixed perovskite film was spin-coated on P3CT-Cs layer at 1000 rpm for 10 s and 4000 rpm for 50 s. 300 μ l chlorobenzene was in-situ dripped onto the perovskite film after 45 s during the second step within 1 s. Afterwards, the perovskite films were immediately transferred to the hotplate and were annealed at 130 $^{\circ}$ C for 7 min. The ambient temperature during film formation is controlled at 25 $^{\circ}$ C to prevent excessively rapid crystallization of the perovskite. To the perovskite films with SnF₂ additive, post-treatment with 0.1 mM EDA in chlorobenzene at 5000 rpm for 20 s is conducted. To the perovskite films without SnF₂ additive, post-treatment with 1.0 mg/ml PbF₂ in isopropanol at 5000 rpm for 20 s is conducted. The post-treatment with PbCl₂ and PbBr₂ is conducted by evaporating 1 nm film on perovskite. Then, a further treatment at 100 $^{\circ}$ C was carried out for 5 min. Finally, 30 nm C60, 6 nm BCP, 1 nm LiF and 100 nm Cu electrode were evaporated under high vacuum ($< 3 \times 10^{-4}$ Torr). The device area defined by the mask was 0.0836 cm².

2. Did the authors investigate the effect of varying PbF₂ concentrations? If so, please include the relevant data and discuss the optimal concentration range.

Response:

We thank the reviewer for this valuable comment. We evaluated the impact of varying PbF₂ concentrations on device performance post-treatment. In this revised version, we have included the corresponding figures and device parameter. Among the tested concentrations, post-treatment with 0.5 mg/ml PbF₂ effectively improved the V_{oc} and FF of the device. The 1 mg/ml PbF₂ post-treatment is identified as the optimal condition, as it yielded the highest V_{oc} (0.884 V) and FF (81.3%), leading to the maximum PCE (24.07%). However, when the concentration was increased to 1.5 mg/ml, a notable decline in FF (67.6%) and PCE (19.59%) is observed, which may be attributed to the obstruction of charge transport caused by excessive PbF₂.

Supplementary Figure 25. The J - V curves of SnF₂-free devices post-treated with different concentrations of PbF₂.

Supplementary Table 4. Summary of photovoltaic performance of SnF₂-free devices post-treated with different concentrations of PbF₂.

PbF ₂ conc. (mg/ml)	V_{oc} (V)	J_{sc} (mA/cm ²)	FF (%)	PCE (%)
0	0.730	32.1	69.5	16.40
0.5	0.860	33.6	74.8	21.66
1.0	0.884	33.5	81.3	24.07
1.5	0.879	32.9	67.6	19.59

3. The manuscript lacks information regarding the morphology of the perovskite films after PbF_2 post-treatment, both in their fresh and aged states. Please provide SEM or AFM images and related analysis.

Response:

We thank the reviewer for this valuable comment. We have included SEM images of Sn-Pb perovskite films without SnF_2 additive and with PbF_2 post-treatment in the Supplementary Fig. 18 to illustrate their morphology before and after aging under 85°C light soaking conditions. The fresh PbF_2 post-treated perovskite films exhibit a morphology that closely resembles that of the fresh perovskite films without SnF_2 additive, as shown in Supplementary Figure 2. Upon aging under 85°C light soaking, the morphology of the PbF_2 post-treated perovskite films remains nearly unchanged, mirroring the aged perovskite films without SnF_2 additive depicted in Figure 1d. These observations suggest that the lack of chemical reaction between PbF_2 and the perovskite components effectively prevents the deterioration of the perovskite morphology.

Supplementary Figure 18. SEM spectra of Sn-Pb perovskite film without SnF_2 additive and with PbF_2 post-treatment before and after aging at 85°C with light soaking for 200 h.

4. The authors claim that the by-product HF reacts with the ITO electrode. However, there is currently insufficient evidence to substantiate this. Please provide experimental data to support this assertion.

Response:

We thank the reviewer for this valuable comment. To further clarify the corrosive impacts of the reaction by-products between FAI and SnF₂ on the ITO substrate, an ITO sample is submerged in a mixed solution of SnF₂ and FAI (0.1 M) for 48 hours and then washed with DMF solvent. The SEM was utilized to characterize the morphology of the corroded ITO surface. The results show a significant morphological change in the ITO substrate when compared to its initial condition. Specifically, the ITO particles on the aged substrates have experienced structural deterioration, losing their original characteristics and taking on a more spherical shape, along with the appearance of conspicuous inter-particle gaps. These findings offer a directly proof that the reaction products of SnF₂ and FAI are harmful to the integrity of the ITO substrate. In the revised manuscript, this result has been included in Supplementary Fig. 9.

Supplementary Figure 9. SEM of ITO substrate before and after 48 h immersion in a 0.1 M SnF₂/FAI mixed solution followed by DMF rinsing.

5. While the experimental results clearly show the superiority of PbF₂ in improving the performance of SnF₂-free Sn-Pb PSCs, a more systematic explanation is needed. The authors should provide further clarification through theoretical analysis or defect characterization techniques.

Response:

We thank the reviewer for this insightful comment. PbF₂ post-treatment effectively lowers the surface Sn/Pb ratio from 1.17 to 0.78 (top-view SEM-EDS), suppressing the tin vacancies (V_{Sn}) formation by replacing Sn²⁺ with Pb²⁺, whose 5s² states lie deeper and form more robust Pb-I bonds than the readily cleaved Sn-I bonds (*Adv. Mater.* **2019**, *31*, 1803792.). In addition, PbF₂ remains chemically inert toward FAI during both fabrication and operation, suppressing A-site vacancies (V_{FA}). Both V_{FA} and V_{Sn} defects are p-type defects that can pin the Fermi level (E_{F}) near the valence band (VB) edge: V_{FA} introduces shallow, delocalized states, while V_{Sn} generates deep, localized states that intensify non-radiative recombination. High concentration of V_{Sn} create a locally iodine-rich environment, while V_{FA} lowers the I⁻ ion migration barrier by reducing steric hindrance (Supplementary Fig. 6), thereby facilitating the formation of further iodine-related defects p-type defects (such as I_{Sn} and I_{FA}). Consequently, these surface p-doping defects are mutually reinforcing (*ACS Energy Lett.* **2024**, *9*, 400-409). Their accumulation bends the bands upward at the perovskite/C60 interface, degrading the quasi-Fermi-level splitting (QFLS). By mitigating such p-type defects, PbF₂ post-treatment enhances interfacial charge transport in p-i-n Sn-Pb PSCs and improves overall device performance. We also supplement this interpretation on page 8-9 in the manuscript.

Supplementary Table 1. The Pb and Sn atomic concentration and Sn/Pb atomic ration obtained in SEM-EDS mapping.

Sample	Pb (At %)	Sn (At %)	Sn/Pb
Pristine	5.7	6.7	1.17
PbF ₂ Post-treatment	5.5	4.4	0.78

Supplementary Figure 14. The Energy Dispersive X-Ray Spectroscopy maps of perovskite films: (a)pristine and (b) after PbF₂ post-treatment.

Supplementary Figure 8. (c) UPS spectra of Sn-Pb perovskite without SnF₂ additive before and after aging at 85°C with light soaking for 200 h.

Supplementary Figure 20. The UPS spectra of the perovskite with SnF₂ post-treatment (a) and PbF₂ post-treatment (b).

6. Can PbF₂ be used as a dopant in the precursor solution instead of SnF₂? If so, how does this affect device performance? Please include a comparative study.

Response:

We thank the reviewer for this valuable comment. The resistance of oxidation of SnF₂ and PbF₂ when used as additives in the tin-lead perovskite precursor solution is compared. As show in the UV-vis absorption spectra of the pristine SnI₂ solution is orange, indicating the presence of SnI₄ (Supplementary Fig. 31). Upon introducing 10 mol% SnF₂ (based on the molar ratio to SnI₂), the solution changes from orange to yellow. This color change occurs because F⁻ has a stronger affinity of for Sn⁴⁺ than Sn²⁺, leading to the selective formation of SnF₄ complexes (*Angew. Chem. Int. Ed.* **2021**, *60*, 21583). SnF₄ has a lower tendency than SnI₄ to incorporate into the perovskite structure,

thereby preventing the formation of tin vacancy defects and improving device performance. In contrast, when PbF_2 is introduced, the strong bonding between Pb^{2+} and F^- makes it difficult for F^- to complex with Sn^{4+} . As a result, PbF_2 is less effective than SnF_2 in preventing Sn^{4+} from being incorporated into the perovskite structure. Consequently, after adding into 10 mol% PbF_2 (based on the molar ratio to SnI_2), the color of the solution remains almost unchanged (Supplementary Fig. 31). The device performance with PbF_2 as an additive is also less competitive as that with SnF_2 additive, as demonstrated in Supplementary Fig. 32 and Supplementary Tab. 9. We also supplemented this data and discussion in the manuscript and Supplementary Materials.

Supplementary Figure 31. The UV-vis absorption spectra of tin-lead perovskite precursor solution with SnF_2 (a) and PbF_2 (b) additive aging in air. (c) The color changes of SnI_2 with SnF_2 and PbF_2 additive.

Supplementary Figure 32. J - V curves of Sn-Pb PSCs with different concentration PbF_2 additive.

Supplementary Table 9. Summary of photovoltaic performance of Sn-Pb perovskite PSCs with different concentration PbF_2 additive.

PbF_2 conc. (mol% vs. perovskite)	V_{oc} (V)	J_{sc} (mA/cm ²)	FF (%)	PCE (%)
0	0.73	32.1	69.5	16.4
2.5	0.866	31.1	62.9	17.02
5.0	0.885	32.6	66.1	19.1
7.5	0.885	34.6	51.7	15.08

7. The use of Pb powder in the precursor solution, rather than Sn powder, is unusual and may raise questions for the reader. Please clarify the rationale behind this choice.

Response:

We thank the reviewer for this valuable suggestion. The primary distinction between Pb and Sn powders lies in their reaction rates with FAI. When Pb or Sn powders react with FAI, they produce FA and their respective metal iodides, PbI_2 or SnI_2 , both of which exhibit a yellow color in DMF solvent. Pb powder reacts rapidly with FAI in the perovskite precursors, as evidenced by the solution quickly turning yellow upon the addition of Pb powder to the FAI solution (Supplementary Fig. 10). In contrast, Sn powder reacts more slowly with FAI, and no significant color change is observed after adding Sn powder to the FAI solution. These results indicate that Pb powder can react more completely with FAI during the perovskite precursor preparation process, generating a greater amount of FA. This increased FA production helps regulate the crystallization growth of perovskite films (*Joule*, **2021**, *5*, 2904-2914), as demonstrated by the SEM images (Supplementary Fig. 10).

Supplementary Figure 10. (a) The UV-vis absorption spectra of FAI solution with lead powder or tin powder. (b-d) The SEM of Sn-Pb perovskite film derived from precursors with (b) lead powder, (c) tin powder and (d) no metal powder (control).

Reviewer #4:

The manuscript entitled “A Tin Fluoride-Free, Efficient and Durable Tin-Lead Perovskite Solar Cell” reveals that the instability of SnF₂ in Sn-Pb perovskite solar cells originates from its thermal reaction with FAI, which accelerates device degradation. The authors propose an innovative strategy by introducing lead powder in the precursor solution combined with a PbF₂ post-treatment to replace the role of SnF₂ in film formation and surface defect passivation. This approach effectively balances efficiency and stability. As a result, the power conversion efficiency of SnF₂-free devices is improved from 16.43% to 24.07%, with excellent photo-thermal stability—maintaining 60% of the initial efficiency after 550 hours of operation at 85°C under MPP conditions. The study is well-structured, the data are comprehensive, and the findings are of significant importance for the development of both single-junction and all-perovskite tandem solar cells. However, several issues should be thoroughly addressed and resolved prior to publication.

Response:

We would also like to thank the reviewer for the valuable comments and helpful suggestions. In the following pages, we provide point-by-point responses to all the questions raised by the reviewers. The corrections in the manuscript are shown in yellow for convenience. We also carefully read the entire manuscript and revised any errors we found. We hope that we have appropriately addressed all the mentioned issues.

1、 The authors identified that SnF₂, when used as an additive, presents issues related to thermal stability, and they accordingly employed metallic Pb to suppress oxidation and used PbF₂ post-treatment for defect passivation. However, the current evidence supporting the instability caused by SnF₂ relies mainly on TG analysis. To strengthen the conclusion, the authors are encouraged to provide further theoretical explanations and complementary characterizations to support this claim. Additionally, it would be valuable to discuss whether similar findings have been reported in the literature.

Response:

We thank the reviewer for this valuable suggestion. To further elucidate the reaction between FAI and SnF₂ from multiple perspectives, we have incorporated a series of Nuclear Magnetic Resonance (NMR) characterizations. In the ¹H NMR spectrum of FAI, the primary characteristic signals of FA⁺ are observed at 7.8 ppm (corresponding to the CH group) and 8.7 ppm (assigned to the NH₂ group) (Supplementary Fig. 2). Upon the addition of SnF₂ to the FAI solution, a noticeable splitting pattern of the NH₂ signal is observed, indicating a strong interaction between the F⁻ ions in SnF₂ and the NH₂ group in FA⁺.

Supplementary Figure 2. The full ^1H NMR spectra of (a) FAI, (b) FAI+SnF₂ and (c) FAI+SnF₂ aging at 85 °C for 3 hours.

Following the aging of the FAI-SnF₂ mixture at 85 °C for 3 hours, a new signal peak emerges at 9.3 ppm. This peak can be attributed to triazine, a known by-product formed during the cyclization of FA molecules after the deprotonation of FA⁺ (Supplementary Fig. 2) (*Sol. RRL* **2021**, 5, 2000715). Unlikely, no significant peak deformation or splitting of the NH₂ signal is observed in the FAI-PbF₂ mixture, suggesting a weaker interaction between FAI and PbF₂. Moreover, no new peak at 9.3 ppm was detected after heating the FAI-PbF₂ sample at 85 °C for 3 hours (Supplementary Fig. 16). These findings further confirm the enhanced reactivity of

SnF₂ compared to PbF₂ in this context. This observation chemically supports the reaction mechanism described in our study. The newly obtained data have been added to Supplementary Fig. 2 and Supplementary Fig. 16 in the revised manuscript. The reaction between FAI and SnF₂ under thermal stress and its influence to the thermal stability of Sn-Pb perovskite film and device are firstly clarified in this work.

Supplementary Figure 16. The full ¹H NMR spectra of (a) FAI+PbF₂ and (b) FAI+PbF₂ aging at 85°C for 3 hours.

2、 It is recommended that the authors include additional performance data for devices fabricated using PbF_2 as an additive, in order to better justify the rationale for employing PbF_2 as a post-treatment rather than incorporating it directly into the precursor. Moreover, while the authors emphasize the synergistic effect of Pb and PbF_2 , data showing the individual contributions of each component are lacking. Further clarification is needed regarding the combined effect of SnF_2 and PbF_2 , and whether their interaction is beneficial or detrimental to device performance.

Response:

We thank the reviewer for this valuable suggestion.

(1) The device performance when using PbF_2 as an additive. In accordance with the reviewers' recommendations, we conducted an investigation into the device performance when PbF_2 is used as an additive (Supplementary Fig. 32 and Supplementary Tab. 9). Regrettably, the device performance upon PbF_2 doping is inferior. Among all the cases, the optimal doping concentration is 5%, resulting in a PCE of 19.1%. This experiment suggests that PbF_2 is more appropriately employed as a post-treatment agent rather than an additive in this context.

Supplementary Figure 32. J - V curves of Sn-Pb PSCs with different concentration PbF_2 additive.

Supplementary Table 9. Summary of photovoltaic performance of Sn-Pb perovskite PSCs with different concentration PbF₂ additive.

PbF ₂ conc. (mol% vs. perovskite)	V_{oc} (V)	J_{sc} (mA/cm ²)	FF (%)	PCE (%)
0	0.73	32.1	69.5	16.4
2.5	0.866	31.1	62.9	17.02
5.0	0.885	32.6	66.1	19.1
7.5	0.885	34.6	51.7	15.08

To clarify the poor device performance when using PbF₂ as an additive. We compare the oxidation resistance of SnF₂ and PbF₂ as additives in tin-lead perovskite precursor solutions. As show in the orange color and UV-vis absorption spectra of the pristine SnI₂ solution, indicating the presence of SnI₄ (Supplementary Fig. 31). Adding 10 mol% SnF₂ (relative to SnI₂ molar ratio) turns the solution from orange to yellow. This color change arises because F⁻ has a stronger affinity for Sn⁴⁺ than Sn²⁺, enabling selective formation of SnF₄ complexes (*Angew. Chem. Int. Ed.* **2021**, 60, 21583). Compared to SnI₄, SnF₄ has a lower tendency to incorporate into the perovskite structure, thus preventing tin vacancy defects and enhancing device performance. In contrast, PbF₂ forms strong bonds with Pb²⁺, making it harder for F⁻ to complex with Sn⁴⁺. Consequently, PbF₂ is less effective than SnF₂ at inhibiting Sn⁴⁺ incorporation into the perovskite structure.

Supplementary Figure 31. The UV-vis absorption spectra of tin-lead perovskite precursor solution with SnF_2 (a) and PbF_2 (b) additive aging in air. (c) The color changes of SnI_2 with SnF_2 and PbF_2 additive.

(2) The effect of Pb powder. In our experiment, to substitute the reducing and crystallization adjustment effect of SnF_2 in precursor, Pb powder is used. Previous studies from our group have demonstrated that Pb has a suitable redox potential to reduce Sn^{4+} without affecting Sn^{2+} (Joule, 2021, 5, 2904-2914). Additionally, Pb powder reacts rapidly with FAI in the perovskite precursors, as evidenced by the solution quickly turning yellow upon the addition of Pb powder to the FAI solution (Supplementary Fig. 10). This reaction produces FA which helps regulate the

crystallization growth of perovskite films, as demonstrated by the SEM images (Supplementary Fig. 10).

Supplementary Figure 10. (a) The UV-vis absorption spectra of FAI solution with lead powder or tin powder. (b-d) The SEM of Sn-Pb perovskite film derived from precursors with (b) lead powder, (c) tin powder and (d) no metal powder (control).

(3) **The effect of PbF_2 post-treatment.** PbF_2 remains chemically inert toward FAI during both fabrication and operation, suppressing A-site vacancies (V_{FA}). Both V_{FA} and V_{Sn} defects are p-type defects that can pin the Fermi level (E_{F}) near the valence band (VB) edge: V_{FA} introduces shallow, delocalized states, while V_{Sn} generates deep, localized states that intensify non-radiative recombination. High concentration of V_{Sn} create a locally iodine-rich environment, while V_{FA} lowers the I^- ion migration barrier by reducing steric hindrance, thereby facilitating the formation of further iodine-related defects p-type defects (such as I_{Sn} and I_{FA}). Consequently, these surface p-doping defects are mutually reinforcing (*ACS Energy Lett.* **2024**, *9*, 400-409). Their accumulation bends the bands upward at the perovskite/ C_{60} interface, degrading the

quasi-Fermi-level splitting (QFLS). By mitigating such p-type defects, PbF₂ post-treatment enhances interfacial charge transport in p-i-n Sn-Pb PSCs and improves overall device performance.

Supplementary Figure 8. (c) UPS spectra of Sn-Pb perovskite without SnF₂ additive before and after aging at 85°C with light soaking for 200 h.

Supplementary Figure 20. The UPS spectra of the perovskite with SnF₂ post-treatment (a) and PbF₂ post-treatment (b).

Finally, to evaluate the synergistic effect of SnF₂ additive and PbF₂ post-treatment, we fabricated PSCs incorporating SnF₂ as a dopant and applying PbF₂ post-treatment. This synergistic strategy decreases the PCE of devices to 20.35%. According to previous reports, SnF₂ tends to accumulate at the top and bottom surfaces of the perovskite film (*Adv. Energy Mater.* **2021**, 11, 2101045). Consequently, this synergistic strategy results in an excessive accumulation of PbF₂ and SnF₂ at the interface, leading to over-passivation and a subsequent reduction in PCE.

Supplementary Figure 30. The J - V curves of Sn-Pb PSCs with both SnF₂ additive and PbF₂ post-treatments or only PbF₂ post-treatment.

Supplementary Table 8. Photovoltaic performance of Sn-Pb perovskite with both SnF₂ additive and PbF₂ post-treatments or only PbF₂ post-treatment.

	V_{oc} (V)	J_{sc} (mA/cm ²)	FF (%)	PCE (%)
SnF ₂ additive	0.846	33.8	71.0	20.35
PbF ₂ treatment				
PbF ₂ post-treatment	0.884	33.5	81.3	24.07

3. Figure 1c reveals pronounced pinholes/cracks in the perovskite film attributed to SnF₂ incorporation. Please provide a mechanistic explanation for this morphological degradation, addressing how SnF₂ influences crystallization kinetics or induces lattice strain.

Response:

We thank the reviewer for this valuable comment. The morphological degradation of the perovskite film with SnF₂ additive in Figure 1c is attributed to the reaction between SnF₂ and FAI component in perovskite under photo-thermal stress as verified in TGA-MS results in Figure 1b. The reaction induces the decomposition of perovskite and the reaction products such as FA and HF volatilize under photo-thermal stress, which is the origin of pinholes/cracks in the perovskite film with SnF₂ in Figure 1c. This morphological degradation has no concern with the crystallization quality of perovskite film. Also, as shown in Supplementary Fig. 2a-b, the perovskite film with and without SnF₂ show nearly no morphology difference in addition to the SnF₂ aggradation at grain boundaries due to the crystallization regulation of Pb powder (*Joule*, **2021**, *5*, 2904-2914). Certainly, SnF₂ could regulate the crystal growth orientation inside perovskite film and facilitate the fabrication of highly ordered film. Meanwhile, the addition of SnF₂ reduces the formation of Sn vacancies in Sn-Pb perovskite film indicating by the slight shift of the position of (110) peak to the left (*Adv. Energy Mater.* **2021**, *11*, 2101045). In order to compare the influences of SnF₂ and lead powder on the crystallization kinetics and lattice strain in the perovskite film, we characterize the cross-SEM and XRD of perovskite film.

Supplementary Figure 12. The XRD patterns of Sn-Pb perovskite film in the absence of Pb powder or SnF₂ additive.

SnF₂ doping exerts a significant influence on the crystallization behavior of the film. The observed shift in the XRD signal peak suggests the occurrence of lattice distortion. In contrast, when Pb powder is used, the addition of SnF₂ has minimal effect on the positions of the XRD peaks. It is proposed that the peak shift originates from lattice distortion caused by the presence of Sn⁴⁺, whereas Pb powder effectively removes Sn⁴⁺ from the system. Consequently, SnF₂ doping has a comparatively smaller impact on the crystallization quality of the film in this condition.

Furthermore, cross-sectional SEM images of the films are obtained. For films fabricated without lead powder and SnF₂ additive, distinct bilayer grain structures are evident. This issue is notably mitigated upon SnF₂ doping. On the other hand, when Pb

powder is used, the films exhibited favorable vertical crystallinity irrespective of SnF₂ addition. The corresponding data have been included in Supplementary Fig. 11 of the revised manuscript.

Supplementary Figure 11. The cross-SEM of Sn-Pb perovskite film in the absence of Pb powder or SnF₂ additive.

4、The textual annotations within the device architecture diagram are unclear in scheme 1 and figure 3a. It is recommended to use more distinct and contrasting labels or visual indicators to clearly differentiate the various layers and components of the solar cell structure. In figure S7 and S12, the “Aged 200 h” curves also should be clarified.

Response:

We thank the reviewer for this valuable suggestion. The textual annotations within the device architecture diagram in scheme 1 and figure 3a have been changed.

The “Aged 200 h” curves in figure S7 and S12 have also been clarified as “85°C, light 200 h”:

5、 On page 4, the authors report a 25.1% weight loss for the FAI + SnF₂ powders around 500 °C; however, this value is inconsistent with the 25.9% shown in Figure 1a. The authors should clarify this discrepancy to ensure the accuracy and consistency of the data presented.

Response:

We thank the reviewer for this valuable observation. Thermogravimetric analysis of FAI + SnF₂ powders shows a minor fluctuation in weight loss, from 25.0 % at 500 °C to 26.2 % at 700 °C, which falls within experimental error. To ensure the accuracy and consistency of the data presented, we have revised the text to state “the final weight loss of FAI+SnF₂ powders is 25.9%, matching the calculated weight fraction of FA and HF in FAI+SnF₂ powders (25.9%)” . The corresponding marker in Fig. 1a has also been repositioned accordingly.

Figure 1. The reaction between SnF₂ and FAI in perovskite film. (a) TGA heating curves expressed as weight % as a function of applied temperature. (b) TGA-MS results of FASn_{0.5}Pb_{0.5}I₃ perovskite precursor powders with (solid circle) and without (hollow circle) SnF₂ additive.

6、 On page 9, Figure 3d should clearly indicate the conditions corresponding to each colored column.

Response:

We thank the reviewer for this valuable suggestion. The legends corresponding to each colored column in Fig. 3d have been added to clearly indicate their conditions.

Figure 3d. PLQY of the control, target and SnF₂ doped Sn-Pb perovskite films, with and without C60 deposition.

7、 On page 10, the authors state, “The external quantum efficiency (EQE) spectrum of the target device, shown in Figure 4c, exhibits an integrated short-circuit current density (J_{sc}) of 32.55 mA cm⁻², confirming the consistency of the integrated current density from J - V characteristic.” However, this statement contains two errors: the EQE data are actually shown in Figure 4b, not 4c, and Figure 4c is not referenced or described elsewhere in the manuscript. The authors should revise this sentence to correct the figure number and ensure consistency with the actual content.

Response:

We thank the reviewer for this valuable observation. “Figure 4c” has been corrected to “Figure 4b” in the EQE data description. The description of Figure 4c has also been added in the manuscript as “Figure 4c and Supplementary Fig. 27 present the statistical parameters of 25 control and 25 target devices selected by random sampling.

The narrow distributions and small standard deviations underscore the excellent reproducibility. All target devices exhibit simultaneous gains in V_{oc} and FF, yielding a marked and consistent PCE enhancement after PbF_2 post-treatment, which firmly corroborates the reliability of this interfacial strategy.”